# Structural roles of guide RNAs in the nuclease activity of Cas9 endonuclease

Youngbin Lim[1], So Young Bak[2], Keewon Sung[2], Euihwan Jeong[2], Seung Hwan Lee[3], Jin-Soo Kim[2,3], Sangsu Bae[2,3,†] & Seong Keun Kim[1,2]

The type II CRISPR-associated protein Cas9 recognizes and cleaves target DNA with the help of two guide RNAs (gRNAs; tracrRNA and crRNA). However, the detailed mechanisms and kinetics of these gRNAs in the Cas9 nuclease activity are unclear. Here, we investigate the structural roles of gRNAs in the CRISPR-Cas9 system by single-molecule spectroscopy and reveal a new conformation of inactive Cas9 that is thermodynamically more preferable than active apo-Cas9. We find that tracrRNA prevents Cas9 from changing into the inactive form and leads to the Cas9:gRNA complex. For the Cas9:gRNA complex, we identify sub-conformations of the RNA–DNA heteroduplex during R-loop expansion. Our single-molecule study indicates that the kinetics of the sub-conformations is controlled by the complementarity between crRNA and target DNA. We conclude that both tracrRNA and crRNA regulate the conformations and kinetics of the Cas9 complex, which are crucial in the DNA cleavage activity of the CRISPR-Cas9 system.

[1] Department of Biophysics and Chemical Biology, Seoul National University, Seoul 151-747, Republic of Korea. [2] Department of Chemistry, Seoul National University, Seoul 151-747, Republic of Korea. [3] Center for Genome Engineering, Institute for Basic Science, Seoul 151-747, Republic of Korea. † Present address: Department of Chemistry, Hanyang University, Seoul 133-791, Republic of Korea. Correspondence and requests for materials should be addressed to S.B. (email: sangsubae@hanyang.ac.kr) or to S.K.K. (email: seongkim@snu.ac.kr).

Most archaea and 40% of bacteria have an adaptive immune system called the clustered regularly interspaced short palindromic repeats (CRISPR)-Cas (CRISPR-associated) complex that provides protection against DNAs of invasive bacteriophages[1–3]. The CRISPR locus consists of a repetitive sequence interspaced by short sequences ('spacers') derived from fragments of invasive foreign genomes (protospacers)[4]. CRISPR RNA (crRNA), which contains individual spacer sequences, recognizes target DNA sites via Watson–Crick base pairing to facilitate targeted double-strand breaks[5]. Among CRISPR-Cas systems, the type II CRISPR-Cas system is widely used for genome editing in cultured cells and various organisms because of its high specificity and ease of use[6]. In the type II CRISPR-Cas system, only one protein, Cas9, serves as an RNA-guided endonuclease utilizing two noncoding RNAs, crRNA and trans-activating crRNA (tracrRNA)[7]. A hybridized duplex of crRNA:tracrRNA binds to Cas9 and induces the cleavage of target DNA sequences containing protospacer-adjacent motif (PAM) at the 3′ end of target DNA[8–10].

Since Cas9 endonuclease derived from *Streptococcus pyogenes* was first applied to genome editing in human cells[11–14], extensive conformational studies of guide RNAs (gRNAs; that is, crRNA and tracrRNA) that form Cas9:gRNA protein–nucleic acid complexes have been performed. The RNA conformations of Cas9:gRNA are important in RNA–DNA base pairing, facilitating the discrimination between the target sites and off-target sites of target DNA. Similar to other nucleic acid-related enzymatic processes, gRNAs lead to multiple conformationally different Cas9:gRNA intermediates with different enzymatic activities. Previous biochemical assays[15] and structural studies using X-ray crystallography[16–18] and single-particle electron microscopy imaging[19] have shown that gRNAs are involved in the expansion of the R-loop structure and initiate large-scale conformational changes of Cas9 for target DNA recognition. However, the mechanistic details of the independent roles of the two gRNAs in Cas9 nuclease activation are still unclear.

The goal of this study is to investigate and unravel the molecular mechanism of the two gRNAs separately on Cas9:gRNA complexation and their conformational changes associated with the nuclease activity. Using single-molecule fluorescence spectroscopy, we find an inactive state of apo-Cas9, transition toward which is suppressed by the tracrRNA–Cas9 interaction. We also find conformational heterogeneity in the crRNA–DNA R-loop structure, whose distribution regulates the nuclease activity of Cas9 endonuclease. These results show that tracrRNA governs the formation of the catalytically active Cas9–gRNA complex, whereas crRNA regulates the nuclease activation of the Cas9–gRNA–DNA ternary complex.

## Results

**Visualization of Cas9-mediated cleavage.** To visualize Cas9-induced DNA cleavage activities on the immobilized target DNA, we used total internal reflection fluorescence microscopy. We designed biotinylated 48-bp double-stranded DNA (dsDNA) dual-labelled with Cy3 and Cy5 at the opposite ends of each strand and tethered to the surface via biotin–neutravidin interaction (Supplementary Fig. 1 and Supplementary Table 1). We pre-incubated 2 nM Cas9 with excess gRNAs (300 nM) for 20 min at 37 °C to form Cas9:gRNAs and then injected the mixture onto the immobilized DNA. Progress of DNA cleavage by Cas9:gRNAs and dissociation of cleaved DNA fragments from the Cas9:gRNAs were monitored by measuring the ratio of the number of Cy5 molecules to that of Cy3 molecules. Immediately

on adding Cas9:gRNAs to the target DNA, we found only minor changes in the number of Cy3 and Cy5 molecules, which is consistent with a previous report in which Cas9:gRNAs were found to remain bound to the DNA even after DNA cleavage[15]. To explicitly observe dissociation of Cas9:gRNAs from the cleaved DNA, we injected 7 M urea solution while rapidly performing a wash to prevent urea-mediated denaturation of neutravidin or DNA double-helix structures. Once Cas9:gRNAs were dissociated from immobilized DNA, the ratio of Cy5 to Cy3 decreased significantly; approximately 80% of target DNAs were cleaved in 5 min (Supplementary Fig. 1).

**Effect of tracrRNA on Cas9 conformation.** To investigate the role of gRNAs in the DNA cleavage reaction, we used a set of partially pre-incubated Cas9:gRNAs to compare the DNA cleavage efficiency. Three different processes of pre-incubation and mixing scheme were used before we injected the mixture onto the immobilized DNA (Fig. 1a): (i) apo-Cas9 and tracrRNA pre-incubated at 37 °C for 20 min, and crRNA added immediately afterwards; (ii) apo-Cas9 and crRNA pre-incubated at 37 °C for 20 min, and tracrRNA added later; and (iii) apo-Cas9 alone pre-incubated at 37 °C for 20 min, and tracrRNA and crRNA added later. In addition, we also used apo-Cas9 pre-incubated at 37 °C for 20 min with both gRNAs (full complex) as a positive control. The cleavage efficiency was then measured at 37 °C after an incubation time of 5 min, the time required to achieve maximum cleavage under our experimental conditions. For Cas9 partially pre-incubated with tracrRNA ((i) in Fig. 1a,b), the efficiency of DNA cleavage (>80% DNA cleaved) was almost the same as that of the 'full complex' (Fig. 1b, green bar). In contrast, Cas9 pre-incubated in the absence of tracrRNA ((ii) and (iii) in Fig. 1a,b) yielded much lower cleavage efficiencies, which suggests that the tracrRNA plays a crucial role in facilitating crRNA binding and DNA cleavage by Cas9.

DNA cleavage by Cas9 pre-incubated without tracrRNA (process (ii) in Fig. 1a) also showed a strong dependence on the pre-incubation temperature, that is, the cleavage efficiency was reduced as the pre-incubation temperature was increased (Fig. 1c, yellow bars). As a control, however, the cleavage efficiency for the direct injection of the mixture with apo-Cas9, crRNA and tracrRNA ('without pre-incubation', Fig. 1c, grey bar) retained nearly 100% of the cleavage activity of the 'full complex'. Because Cas9:gRNAs do not undergo multiple turnover cycles[15], different DNA cleavage efficiencies represent different molecular ratios of catalytically active Cas9 to inactive Cas9. Therefore, as the pre-incubation temperature increases, more active Cas9 species are converted into the inactive conformation in the absence of tracrRNA, possibly through an energy-barrier route. Temperature-dependent circular dichroism (CD) spectra suggest that the energy-consuming transformation from active Cas9 to inactive Cas9 is associated with a conformational change (Fig. 1d). As the pre-incubation temperature increased from 25 °C to 37 °C, a change in the ellipticity of Cas9 was observed in the characteristic band near 210 ∼ 220 nm (from the green curve to the black one). However, when we annealed Cas9 by slowly cooling it from 37 °C to 25 °C, no reverse change was observed (the dashed red curve remaining identical to the green curve), indicating that there was no reverse conformational rearrangement to the active form. Taken together, these results suggest that the conformational structure of inactive Cas9 is thermodynamically more stable and kinetically trapped at lower temperatures owing to the large refolding barrier.

To clarify whether inactive Cas9 interacts with gRNAs to form catalytically active Cas9:gRNA, we first prepared inactive Cas9 by heating as in Fig. 1d and conducted a cleavage assay after

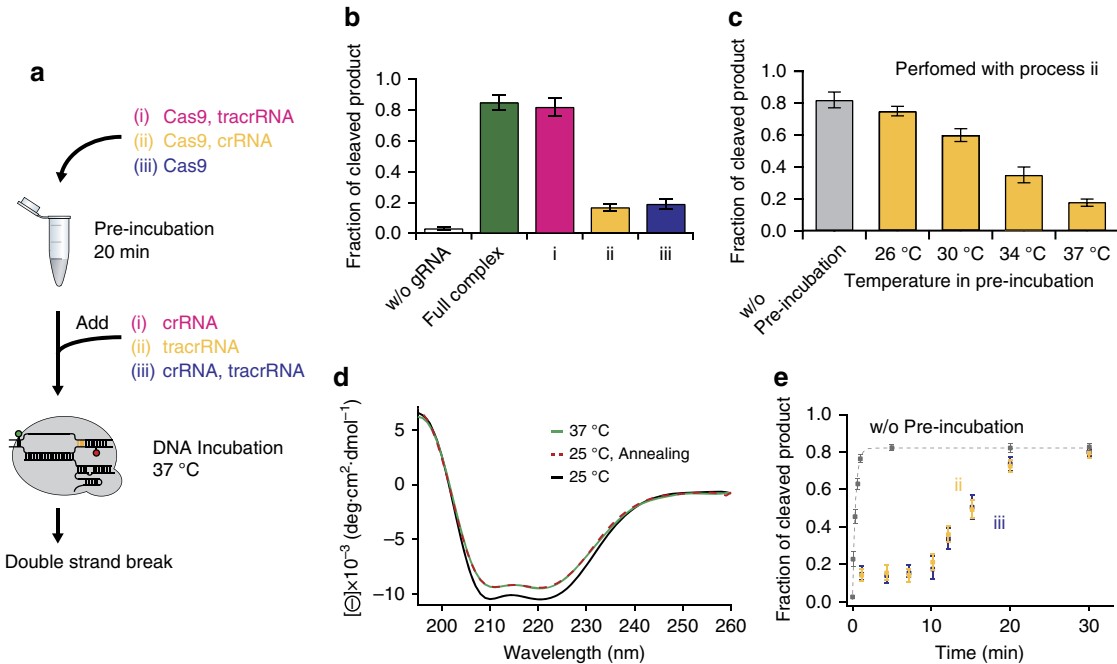

**Figure 1 | Identification and quantification of inactive Cas9.** (**a**) Three different procedures of our partial pre-incubation experiment. (**b,c**) Quantification of DNA cleavage efficiencies under different pre-incubating conditions: mixing process (**b**) or temperature during pre-incubation (**c**; mean ± s.e.m., $n \geq 5$). (**d**) CD spectra of Cas9 without tracrRNA at 25 °C (black) and 37 °C (green). Annealing was performed from 37 °C to 25 °C at a cooling rate of $-1$ °C min$^{-1}$ (red dotted line). (e) Kinetic curves of DNA cleavage in the absence (grey) or presence (yellow: Cas9-crRNA, blue: Cas9 alone) of 37 °C pre-incubation. DNA incubation and fluorescence measurement were conducted at 37 °C for all plots in (**b,c,e**). Error bars in **e** represent s.d.

adequate pre-incubation of inactive Cas9 with two gRNAs ($>1$ h at 25 °C). This experiment yielded a cleavage efficiency of only about 20%, which is similar to those obtained from (ii) and (iii) in Fig. 1b (see also Supplementary Fig. 2). These data are consistent with the aforementioned result showing that inactive Cas9 is trapped at lower temperatures and further support that tracrRNA does not interact with inactive Cas9 or induces reverse conformational changes in inactive Cas9 at least without sufficient thermal energy. To investigate the rearrangement kinetics from inactive Cas9 to catalytically active Cas9 complex at 37 °C, we revisited the cleavage assay with inactive Cas9, using longer DNA incubation times. We pre-incubated inactive Cas9 with crRNA (ii) or no gRNAs (iii) at 37 °C for 20 min, and then incubated them with the immobilized target DNA at 37 °C after adding other gRNAs. The result, given in Fig. 1e, shows that in both cases inactive Cas9 regained the cleavage activity after about 20 min, indicating that the thermal energy at the elevated temperature allows more frequent occurrence of the conformational rearrangement. Kinetically, the recovery of the cleavage efficiency was fully achieved after a lag phase of approximately 10 min under our experimental conditions. The presence of the lag phase indicates that there is a rate-determining step involving reaction intermediates, which could be attributed to the slow conformational rearrangement of inactive Cas9 involving tracrRNA. Finally, these results suggest that the Cas9–tracrRNA interaction shifts the equilibrium away from the inactive conformation, the thermodynamically favourable state in the absence of tracrRNA, by allowing a lower energy state for the active one.

**Conformational heterogeneity of the RNA–DNA heteroduplex.** For enzymatic cleavage, the ternary complex comprising Cas9:gRNA and target DNA (Cas9:gRNA:DNA) must be assembled by the incorporation of target DNA onto Cas9:gRNA.

To elucidate the molecular functions of gRNAs in the formation of the ternary complex, we performed a single-molecule fluorescence resonance energy transfer (FRET) experiment to probe the conformational dynamics of DNA and crRNA during Cas9:gRNA:DNA formation. As shown in Fig. 2a, we designed a FRET pair between biotinylated target DNA internally labelled with Cy3 (donor) at the distal 5′ end of the protospacer positioned at the 25th nucleotide from the PAM site ($+25$) and crRNA end-labelled with Cy5 (acceptor) at the 5′ end, so that we could monitor conformational changes occurring during the RNA–DNA hybridization. We first demonstrated FRET measurement for Cas9:gRNA interacting with the fully matched target DNA (wild type), containing a proper PAM (in our measurement, 5′-GGG-3′) and a protospacer with perfect complementarity to crRNA. When we injected Cy5-labelled Cas9:gRNA onto Cy3-labelled DNA while using an optical excitation scheme for Cy3, FRET signals from Cy5 that represented individual DNA binding events were observed in real time (Supplementary Fig. 3). After sufficient incubation ($>30$ min) of Cas9:gRNA with the immobilized target DNA to reach chemical equilibrium, the FRET efficiency data showed two distinct bound states between Cas9:gRNA and DNA characterized by a high-FRET value (0.83) and a low-FRET value (0.27; Fig. 2b). When Cas9 was replaced with dCas9 ('dead' Cas9, a catalytically inactivated form of Cas9 resulting from the mutations D10A and H840A) in a control experiment, identical FRET states were also observed (Supplementary Fig. 4), suggesting that neither FRET state is related to the cleaved product. Rather, these two different values of the FRET efficiency may represent the structural flexibility associated with an intermediate state in the DNA binding and cleavage process, unrelated to the cleaved product.

Next, we characterized the structural conformation and kinetics of each FRET state. As Cas9:gRNA binds to wild-type DNA, we found that binding events in a single-molecule

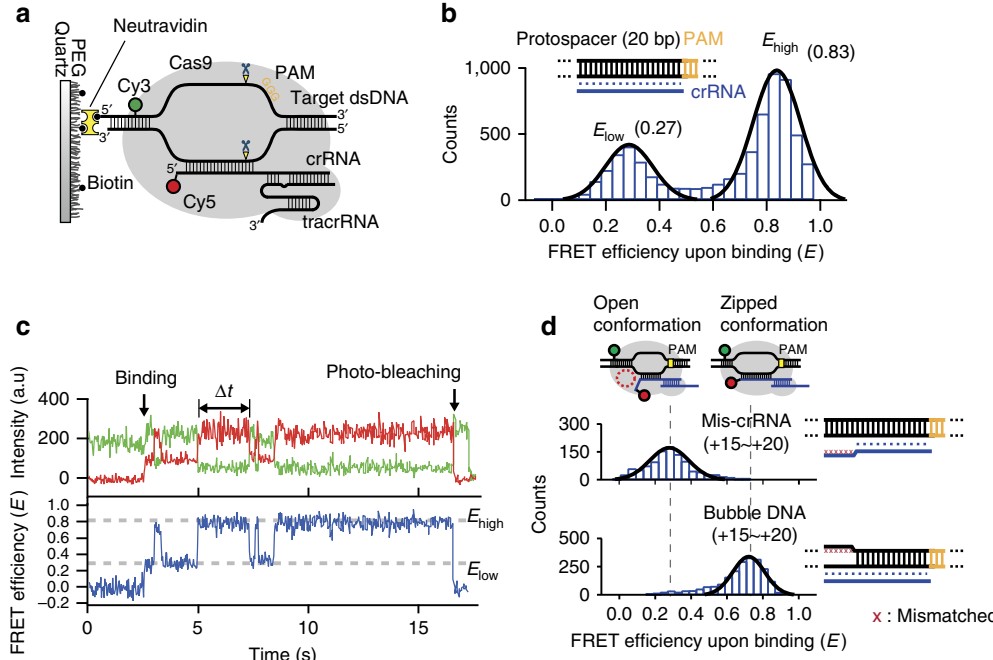

**Figure 2 | Single-molecule FRET analysis for sub-conformation of Cas9:gRNA:DNA.** (**a**) Scheme for smFRET experiment. Cas9 with gRNAs that consist of Cy5 (acceptor)-labelled crRNA, tracrRNA, binds to Cy3 (donor)-labelled dsDNA. In the FRET measurement, a relatively low concentration of gRNAs (30 nM) compared with cleavage experiments was used to reduce the background fluorescence of Cy5-labelled crRNA. (**b**) A histogram of the FRET upon binding (we selected the molecules emitting Cy5 signal) to wild-type target DNA exhibits two peaks centred at 0.27 and 0.83, with Gaussian fits (black line). (**c**) A representative time trajectory representing the short-lived open conformation and zipped conformation exhibits two FRET states. The duration of each conformation is measured as the dwell time ($\Delta t$). The trajectory was imaged right after the injection of Cas9:gRNA into a single-molecule chamber with an integration time of 0.03 s. The points of binding and photo-bleaching were indicated by the black arrows. (**d**) Histograms of FRET upon binding with each PAM-distal mutant exhibit different conformational distribution (with Gaussian fits, black line). The 'mis-crRNA' represents the sequence containing mismatched bases between target and crRNA in PAM-distal region (from $+15$ to $+20$) and the 'bubble DNA' represents the sequence containing mismatched bases between nontarget and target strand in PAM-distal region (from $+15$ to $+20$). In **b** and **d**, diagrammatic representations are used for PAM (yellow), protospacer (black), crRNA (blue) and the approximate location of mismatched bases on target DNA (red cross).

trajectory mostly start in the low-FRET state with the FRET value of 0.27 (Fig. 2c). Moreover, when the target strand of DNA (complementary to the crRNA) and crRNA on the PAM-distal region are mismatched (termed 'mis-crRNA'), the binding events between Cas9:gRNA and target DNA exhibit only a single FRET band centred around the value of 0.27 (Fig. 2d, upper histogram). These results suggest that the low-FRET state in our measurement represents the PAM-proximal bound complex (termed the 'open conformation') in which crRNA partially hybridizes with the dsDNA before the complete formation of the R-loop.

Once the wild type forms the open conformation, as shown in Fig. 2c, the transition to the high FRET state is observed. Given the previously reported X-ray crystal structure of Cas9: gRNA:DNA[16–18], the high-FRET state is thought to represent the conformation after the completion of the R-loop formation (termed the 'zipped conformation'), and the observed FRET change from low-FRET to high-FRET may correspond to the expansion of the R-loop structure. The FRET value of the zipped conformation (0.83) amounts to an inter-probe distance of 40 Å with $R_0 = 56$ Å, which seems reasonable considering the distance between the 5′ end of crRNA and the $+25$ position of the non-target strand of DNA[20]. Interestingly, the R-loop structure does not remain in the high-FRET state, but undergoes a series of transient, repetitive transitions between the two FRET states (whether target DNA is broken or not). The dwell-time distributions of the two FRET states ($E_{high}$ and $E_{low}$) both fit to single exponential functions ($\Delta t(E_{high}) = 222$ ms $\pm 18$ ms and $\Delta t(E_{low}) = 270$ ms $\pm 14$ ms), which are likely to reflect the

R-loop structure kinetics (Supplementary Fig. 5). With the removal of the complementarity in the PAM-distal region (from $+15$ to $+20$) between the two DNA strands (termed 'bubble DNA'), the FRET data (Fig. 2d, bottom histogram) shows only the high-FRET population centred around a FRET value of 0.75 (a notably lower value from that of wild-type DNA, possibly arising from the conformational distortion of the dsDNA by mismatched bases within the protospacer) with no more repetitive transitions. This further supports the idea that the observed FRET transitions represent the relative motions between the target strand of DNA and crRNA at the PAM-distal end. Our results amount to a direct observation of the sub-steps of the R-loop expansion in the ternary complex Cas9:gRNA:DNA, with a clear distinction of the conformational subpopulations of the RNA–DNA heteroduplex.

**crRNA-regulated conformational flexibility of Cas9:gRNA:DNA.** To further monitor the role of crRNA in Cas9:gRNA:DNA, we introduced truncated crRNA in which three nucleotides were omitted in the PAM-distal end. Loss or gain of stability in the interaction between crRNA and the target strand of DNA was expected to cause a subtle perturbation of the conformational equilibrium between the two FRET states. In agreement with our hypothesis, the FRET histogram with truncated crRNA after sufficient incubation time ($>30$ min) exhibited a notably different distribution, where not only the FRET values of the low and high FRET states are different (0.29 and 0.76, respectively) but also their relative populations have been reversed (Fig. 3a), indicating that the

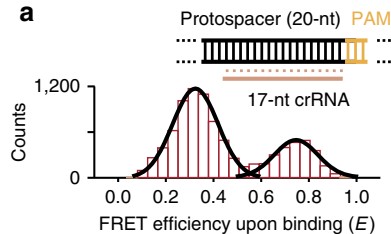

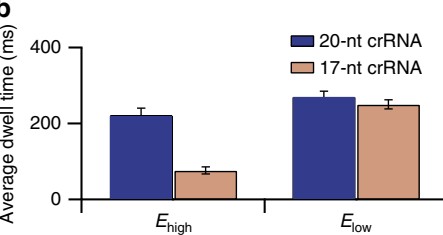

**Figure 3 | Effects of crRNA complementarity on R-loop conformations.** (**a**) A histogram of the FRET upon binding (we selected the molecules emitting Cy5 signal) to wild-type target DNA with 17-nt crRNA exhibits two peaks centred at 0.29 and 0.76, with Gaussian fits (black line). 17-nt crRNA is illustrated by pink line. (**b**) Comparison with each average dwell time of $E_{high}$ and $E_{low}$ from 20-nt or 17-nt crRNA (from dwell time distributions in Supplementary Figs 5 and 6; mean ± s.e.m., $n \geq 3$).

conformational equilibrium at the PAM-distal end is shifted toward base-pairing between the two DNA strands. Considering Cas9:gRNA with 17-nt crRNA exhibited nearly identical FRET values and binding affinity compared with Cas9:gRNA with 20-nt crRNA, we interpret that the shortened nucleotide results in a lower binding energy than the binding energy with 20-nt crRNA at the PAM-distal end of the crRNA–DNA interface, destabilizing the zipped conformation. This result is in good agreement with the previous hypothesis suggesting that truncated crRNAs may exhibit enhanced discrimination against off-target DNA strands because omitted nucleotides reduce the stabilizing energy of the RNA–DNA heteroduplex to a level just sufficient for on-target selectivity[21,22]. Remarkably, in single-molecule trajectories with 17-nt crRNA, the average dwell time of the high-FRET state (77 ms ± 9 ms) was three times shorter than that of the 20-nt crRNA (222 ms ± 18 ms), while the low-FRET dwell time was similar to that of the 20-nt crRNA (Fig. 3b and Supplementary Fig. 6). Our data support that the complementarity of crRNA to the target strand of DNA kinetically influences the conformational transition between the R-loop sub-structures.

**Nuclease activity related to the conformational flexibility.** To determine the relationship between the R-loop sub-structure and the cleavage activity of Cas9, we measured the Cas9 cleavage efficiency with variable dual-labelled DNA constructs (wild-type DNA, bubble DNA and mis-crRNA) having different conformational distributions of RNA–DNA heteroduplex, as measured in single-molecule FRET histograms (Fig. 2b,d). We found that the DNA sequences exhibiting the zipped conformation (wild-type and bubble DNA) were cleaved by the Cas9 complex, while the mis-crRNA, which retained the level of binding affinity but showed only the open conformation, failed to cleave the target DNA (Fig. 4a). Furthermore, a comparison of the kinetics of the wild-type and bubble DNA showed that the greater proportion of time spent in the open conformation resulted in a slower rate of DNA cleavage (Supplementary Fig. 7). Once target DNA is bound by the Cas9 complex, DNA melting in the PAM-distal region of the DNA duplex and formation of the RNA–DNA heteroduplex are the rate-determining steps. These results reveal that the open conformation is incompatible with DNA cleavage and suggest that DNA cleavage events are initiated only if a stable R-loop forms.

To elucidate how the stability of the R-loop structure affects DNA cleavage, we sought to vary the stability of the R-loop structure using off-target DNA sequences with the introduction of mismatched nucleotides at variable positions (Supplementary Table 1). The binding of Cas9:gRNA to target DNA became significantly more stable when at least half of the protospacer sequences upstream from the PAM site were matched; DNA sequences containing a mutation in the 'seed' region did not

exhibit significant DNA binding, as expected (Supplementary Fig. 8)[23]. In agreement with the enhanced cleavage of the bubble substrate, we found that the cleavage efficiency increased as the population of the high-FRET state increased (Fig. 4b). Notably, we found that the mismatch at 10–11 did not exhibit the high-FRET state, despite the complementarity of nine nucleotides at the PAM-distal end, suggesting that the formation of the zipped conformation, in addition to the complementarity of nucleotides at the PAM-distal end, requires the stability of the entire RNA–DNA heteroduplex. These results show that the FRET transition between the two states involves both R-loop expansion as well as the enzymatic activation of Cas9 complex. Therefore, the conformational changes between crRNA and its target strand of DNA serve as a key step leading to activation of the Cas9:gRNA:DNA structure for DNA cleavage.

**Discussion**
Our results demonstrate the roles of both of tracrRNA and crRNA in the regulation of Cas9 complex structures associated with Cas9 nuclease activity. We found that tracrRNA is required to maintain the active conformation of apo-Cas9 for target DNA binding, whereas crRNA regulates the R-loop structure according to the stability of the RNA–DNA heteroduplex (Fig. 5). Several structural studies of type II CRISPR-Cas9 have suggested that Cas9 undergoes conformational changes during gRNA binding and target DNA recognition, but we discovered conformational change taking place in apo-Cas9 itself without nucleic acid interactions although its significance *in vivo* remains to be addressed by further studies. In the absence of tracrRNA, apo-Cas9 becomes inactive, suggesting that tracrRNA (possibly the tracrRNA moiety in the partial tracrRNA:crRNA duplex) is recognized by structurally active apo-Cas9 to form Cas9:gRNA. Consistent with our model, previous biochemical data showed that Cas9–tracrRNA did not exhibit nonspecific binding; nonspecific binding was observed only with Cas9 alone or Cas9–crRNA[10]. These results indicate that the tracrRNA–Cas9 interaction affects the conformation of Cas9 for target DNA recognition.

A previous *in vivo* study showed that Cas9 is also involved in the maturation of the hybridized pre-gRNAs (pre-crRNA and pre-tracrRNA) by RNase III (ref. 5). Our *in vitro* study sheds some light on the role of Cas9 in this process, through which pre-tracrRNA may be recognized by Cas9 and recruited to pre-crRNA with the help of Cas9, similar to the Cas9–gRNA complexation process for target DNA cleavage.

Our real-time single-molecule data demonstrated that the formation of the R-loop structure inside Cas9:gRNA:DNA was a multistep process and exhibited a dynamic motion at the PAM-distal end of the protospacer. The observed conformational transitions (Fig. 3a) are consistent with the previous crystal

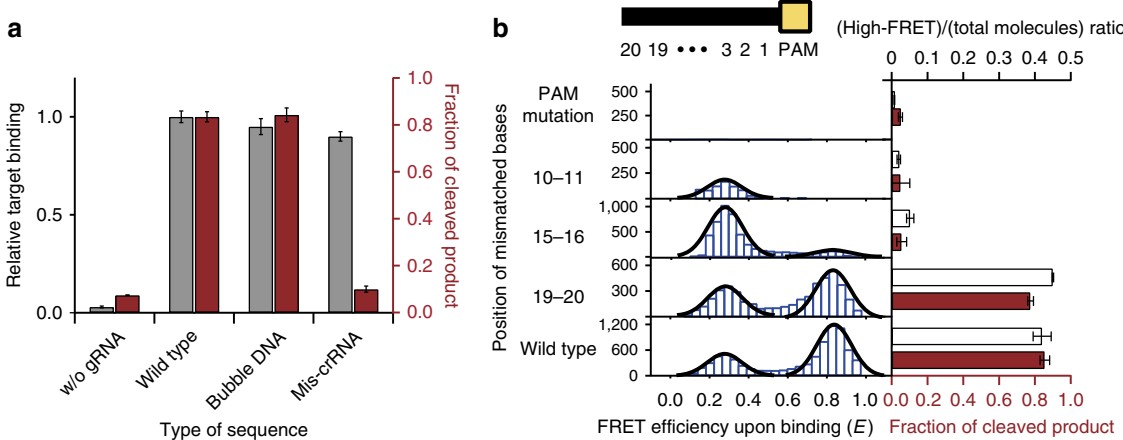

**Figure 4 | Cas9 cleavage activity with various mutated target DNA. (a)** The relative target binding (grey bars) versus the fraction of cleaved product (red bars) for DNA sequences containing various positions of mismatched bases. (mean ± s.e.m., $n \geq 5$). **(b)** Histograms of FRET upon binding with each mismatched mutant with Gaussian fits (black line). The ratio of high-FRET states to all DNA-bound states calculated from the FRET histograms (white bars) versus the DNA cleavage efficiency (red bars) for different DNA sequences containing various positions of mismatched bases. The base positions are labelled by numbering from the 5′ end of PAM as shown on top panel.

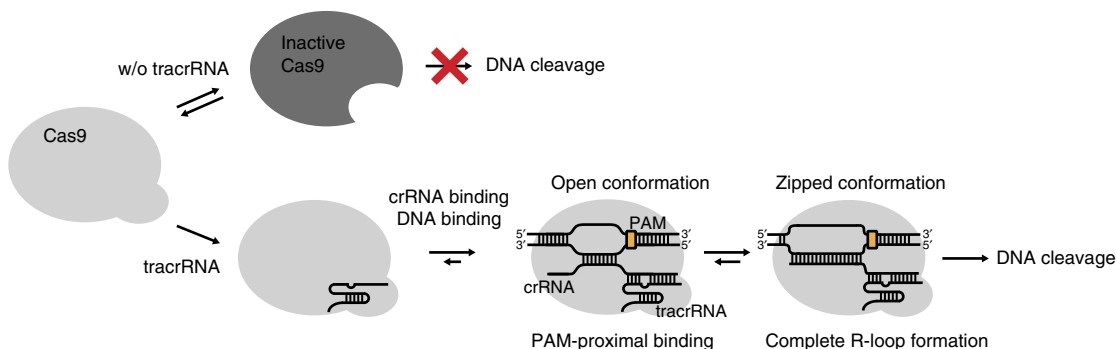

**Figure 5 | Scheme of the conformational roles of both tracrRNA and crRNA during Cas9 nuclease activity.** Model for the conformational regulation of gRNAs during target DNA binding and cleavage process by Cas9:gRNA complex.

structures of Cas9, which show conformational flexibility[17,18], and with recent biophysical studies reporting that Cas9, in contrast to the type I CRISPR Cascade, does not exhibit a 'locking' mode in which Cas9:gRNA:DNA is locked for nuclease activity after the R-loop formation[24]. On the other hand, the rate of the observed transitions (Fig. 2c) was two orders of magnitude faster than the target cleavage rate (Fig. 1e), which suggests that the formation of the R-loop is not sufficient for Cas9-mediated DNA cleavage *per se*, although the cleavage does not occur unless the zipped conformation is formed. As several studies have shown that the nuclease activity of Cas9 is associated with the movement of the HNH nuclease domain that actually cleaves the target strand[17,18,25], it is possible that Cas9 undergoes a conformational change into the zipped conformation that may trigger structural rearrangement of the nuclease domain.

Our experiments with variable off-target sequences demonstrated that the stability of the crRNA–DNA interface was crucial for the conformational distribution and kinetics of Cas9:gRNA:DNA. Off-target sequences that failed to overcome the energetic barrier for the zipped conformation exhibited only PAM-proximally bound structures and were not cleaved (Fig. 4 and Supplementary Fig. 7). Although further experiments are necessary to elucidate the details of DNA cleavage mechanism of Cas9:gRNA:DNA such as synchronous single-molecule measurement for R-loop completion and HNH domain

movement, our results showed that the stability and kinetics of the R-loop structure based on the crRNA–DNA heteroduplex serve as key conformational regulators of the enzymatic behaviour of Cas9. Furthermore, this newly unveiled role of the kinetics and dynamics of the R-loop structure of Cas9:gRNA:DNA could allow prediction of off-target sites quantitatively based on the calculation of the stability of the crRNA–DNA interface, thereby enabling more efficient target design in genome editing.

## Methods

**Cas9 and nucleic acids.** Recombinant Cas9 and dCas9 from S. pyogenes were overexpressed from *Escherichia coli* or purchased from New England Biolabs. In case of expressed proteins, Cas9 gene was subcloned into pET28-b(+). Recombinant Cas9 protein containing a nuclear localization signal, the HA epitope, and the His-tag at the amino (N) terminus was expressed in BL21(DE3) strain, purified using Ni-NTA agarose beads (Qiagen), and dialysed against 20 mM HEPES pH 7.5, 150 mM KCl, 1 mM DTT and 10% glycerol. The purified Cas9 protein was concentrated using Ultracel 100 K cellulose column (Millipore). The purity and concentration of Cas9 protein were analysed by SDS– polyacrylamide gel electrophoresis. We also confirmed that the catalytic activities and DNA binding affinities of the purified Cas9 were comparable to those of the purchased Cas9 upon our experimental conditions.

All DNA oligonucleotides were purchased from Integrated DNA Technologies. Biotin was conjugated at the 5′ end of nontarget strand. Internal amino modification for dye labelling was incorporated in each DNA strand, and both Cy3 and Cy5 were conjugated via a nucleophilic substitution reaction between amino-modified thymine and NHS ester-linked fluorescent dyes.

Guide RNAs were *in vitro* transcribed through run-off reactions by T7 RNA polymerase using the MEGAshortscript T7 kit (Ambion) according to the manufacturer's manual. Transcribed RNA was purified by phenol:chloroform extraction, chloroform extraction and ethanol precipitation. Purified RNA was quantified by spectrometry. We also used guide RNAs based on solid-phase synthesis (purchased from ST Pharm) in single-molecule FRET measurements. Cy5-labelled crRNA was purchased and HPLC-purified from ST Pharm. All RNA-related solutions were prepared with nuclease-free water. Devices and lab space were treated with 70% ethanol, and all disposable tips and microcentrifuge tubes were certified as RNase-free.

DNA/RNA sequences and the positions of biotin and fluorophores are provided in Supplementary Table 1.

**Single-molecule measurement.** Coverslips and quartz glasses were passivated by polyethylene glycol to prevent samples from nonspecific binding on the glass surface[26]. All imaging and cleavage reaction were performed at 37 °C (unless otherwise specified) with the following buffer composition: 100 mM NaCl, 50 mM Tris-HCl pH 7.9, 10 mM MgCl$_2$, 1 mM DTT and 0.1 mg ml$^{-1}$ BSA.

For cleavage and binding measurements and smFRET experiments for FRET histograms, the oxygen scavenger (2.7 U ml$^{-1}$ of pyranose oxidase (Sigma-Aldrich), 7.5 U ml$^{-1}$ of catalase (Sigma-Aldrich) and 0.4% (w/v) of β-D-glucose) and the triplet quencher (2 mM Trolox) were applied to the buffer to prevent the organic fluorophores from severe photo-fatigue[27]. The cleavage efficiencies were measured after 5 min incubation of Cas9:gRNA (2 nM Cas9, 300 nM gRNAs) with DNA unless stated otherwise. The binding affinities and FRET histograms were obtained from the images after 30 min incubation of Cas9:gRNA (2 nM Cas9, 30 nM gRNAs) with DNA.

In case of experiments for single-molecule time traces, imaging was performed at room temperature in the same condition with the aforementioned description except for the oxygen scavenging system (1 mg ml$^{-1}$ of glucose oxidase (Sigma-Aldrich), 0.04 mg ml$^{-1}$ of catalase (Sigma-Aldrich) and 0.8% (w/v) of β-D-glucose) and the addition of 5% (v/v) of glycerol. The time traces were acquired intermittently during the incubation (from 0 min to 30 min) of Cas9:gRNA with DNA.

In all single-molecule measurements, we constructed a flow chamber by assembling a microscope slide and a coverslip with double-sided tape and sealing with epoxy. We adopted rounded-holes on the slide as the inlet and outlet of solution exchange.

**Set-up and data analysis.** We obtained all the experimental data of fluorescence assay from home-built total internal reflection fluorescence microscopy. For smFRET assays, we excited fluorophores with a 532 nm green laser (Samba, Cobolt AB), and for cleavage assays, a 633 nm He–Ne laser (25-LHP-925-230, Melles-Griot) along with the green laser were used to excite Cy3 and Cy5 molecules independently. The emitting photons from Cy3 and Cy5 were collected by a water-immersion objective lens (UPlanApo × 60, Olympus) and then filtered by emission filter (ZET405/488/532/642m, Chroma Technology Corp.). The fluorescence emission signal was split into green and red channels with a dichroic mirror (645dcxr, Chroma Technology Corp.) and detected by back-illuminated electron-multiplying charge-coupled device (iXon DU-897, Andor Technology). All the images and time traces were obtained with a homemade C++ script (available for download at https://cplc.illinois.edu/software) and analysed by MATLAB and IDL programme.

To determine the cleavage ratio of DNA and the bound ratio of Cas9:gRNAs, we quantified the number of Cy5 molecules before and after the injection of the reconstituted Cas9:gRNAs. We used home-built IDL and MATLAB codes to count the number of Gaussian-fitted spots from measured images and then calculated the average of (numbers of Cy3 molecules)/(numbers of Cy5 molecules) after making consideration for photo-bleaching and/or urea-mediated denaturation of the DNA duplex.

In the smFRET histogram and time trace, the FRET efficiency was calculated as the ratio of fluorescence intensities, $\beta I_{acceptor}/(\gamma I_{donor} + \beta I_{acceptor})$. The correction factor $\gamma$ is calculated as the ratio of change in the acceptor intensity, $\Delta I_{acceptor}$, to change in the donor intensity, $\Delta I_{donor}$, upon acceptor photo-bleaching in the absence of a protein ($\gamma = \Delta I_{acceptor}/\Delta I_{donor} = 1$ for Cy3–Cy5 pair in our experimental conditions). The other correction factor $\beta$ was used to account for the enhanced fluorescence intensity of Cy5 owing to changes in its photophysical properties on protein binding. The value of $\beta$ in our system was 0.5. To determine the transition of FRET states and to calculate the transition rates between the states, we used hidden Markov modelling based on variational Bayesian expectation maximization[28]. To determine the high FRET ratio from the FRET histogram, we sorted high- and low-FRET states with an appropriate threshold. In this report, at least 10 measurements were repeated and collected to display all plots and histograms, with each single measurement monitoring over 200 molecules simultaneously.

**CD measurement.** CD spectra of Cas9 protein were measured over the range of 190 to 260 nm with Chirascan plus (Applied Photophysics) at various temperatures. Wave scans were acquired by sampling data at 1 nm intervals between 190 and 260 nm. Temperature was controlled by Peltier-type temperature controller in the range of 25 to 37 °C. CD spectra were obtained using 1 mg ml$^{-1}$ of Cas9 protein in Cas9 storage buffer (20 mM HEPES pH 7.5, 150 mM KCl, 1 mM DTT and 10% glycerol). Each spectrum was obtained from an average of three scans and the result was presented as mean residue ellipticity (deg cm$^2$ dmol$^{-1}$) at each wavelength.

**Data availability.** The data that support the findings of this study are available from the corresponding author on request.

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

## Acknowledgements

This work was supported by the National Research Foundation of Korea through the Global Frontier Program for Bioconvergence (NRF-2014M3A6A4075063). Y.L. acknowledges the support from the Fostering Core Leaders of the Future Basic Science Program (NRF-2011-0002378). This work was also supported by the Institute for Basic Science (IBS-R021-D1) for J.-S.K. and the Plant Molecular Breeding Center of Next Generation BioGreen 21 Program (PJ01119201) grant to S.B. We also acknowledge the BK21 Plus Program and SNU Brain Fusion grant.

## Author contributions

Y.L., S.B., J.-S.K. and S.K.K. designed the research. Y.L., S.Y.B. and K.S. performed the experiments. Y.L., K.S. and S.B. analysed the data. E.J. and S.H.L. prepared the biological samples. S.B. and S.K.K. supervised the research.

## Additional information

**Competing financial interests:** The authors declare no competing financial interests.

**How to cite this article**: Lim, Y. *et al.* Structural roles of guide RNAs in the nuclease activity of Cas9 endonuclease. *Nat. Commun.* **7**, 13350 doi: 10.1038/ncomms13350 (2016).

**Publisher's note**: 

