## [Peer Review File · Nature Communications]

Reviewer #1 (Remarks to the Author):

I have attached my review as a .pdf file.

Reviewer #2 (Remarks to the Author):

Using single molecule FRET and TIRF imaging, Lim et al claim to have characterized two conformational transitions in Cas9/gRNA complexes controlled by the RNAs. In their model, tracrRNA promotes the formation of an "active" conformation early in the assembly pathway, which they demonstrated indirectly using kinetic analysis. The crRNA controls/promotes a transition between "open" (initial) and "zipped" (cleavage active) states of the Cas9/gRNA/targetDNA complex after binding, which they demonstrated using single molecule FRET between the 5'-end (PAM-distal) of the crRNA and the non-hybrid DNA target strand.

The data suggest that in the absence of tracrRNA, Cas9 transitions (on the minute time scale at 37°C) to an inactive conformation that can be induced to slowly reactivate (also on the minute time scale) by tracrRNA. The crRNA-induced transitions occur on the second time scale and are modulated by RNA-DNA complementarity at the PAM-distal end of the heteroduplex.

The authors posit that the crRNA-induced conformational change coincides with R-loop expansion from an initial binding state that samples the PAM-proximal end of the target DNA to the cleavage competent state where the entire crRNA is hybridized with the target strand. They suggest that the time distribution spent between the open and zipped states is determined by PAM-distal complementarity which in turn controls cleavage rate and target specificity.

This paper is the first to investigate in real time conformational changes on Cas9 assembly and target selection. While earlier single molecule work identified distinct PAM-proximal and PAM-distal dependent binding steps and postulated R-loop expansion as the associated structural change (Szczelkun et al 2014), this work provides data that support this connection, provides equilibrium distributions and time scales, and provides direct evidence linking this change to downstream cleavage. Likewise, open and zipped conformations were postulated from crystal structures and provided support for interpreting the FRET data.

Understanding the role of tracrRNA in active complex assembly might be useful in maximizing in vitro assembly reactions or trouble-shooting in vivo ones. The crRNA controlled transition provides mechanistic insights into how Cas9/grRNA sample dsDNA for target site selection.

This work should be of interest to researchers dissecting the structural and kinetic mechanisms of target site selection could certainly, as the authors suggest, lead to design of enhanced specificity variants.

The claims are convincing given the coincidence of the FRET data and other previously published works. The method is internally controlled as all measurements were performed in the context of the TIRF set-up.

Experiments that would strengthen these claims are those that would further test the predictions of their model (observing FRET behavior of Cas9 mutations designed to stabilize one form or the other, and how these affect specificity) seem beyond the scope of a Communications format. The authors appear to have covered most of the relevant literature (although I am not a Cas9 "expert"). One recent publication they might cite and/or discuss involves enhanced specificity mutants gained by destabilizing the PAM-proximal interactions on the non-cleaving strand which retain high cleavage activity even though they bind less well (Slaymaker et al Science 351, 84 (2016)).

Another addition which might be useful, is quantification of transition rate constants for the tracrRNA-dependent slow step as well as rate constants for the cleavage. It is also possible to examine the ΔH via a temperature-dependence series, but this would be some work and perhaps beyond what is required for a Communications paper.

There are some minor scientific and writing issues:

1) The authors might want to address these scientific points:

Page 7 (bottom) "Rather, these two different values of the FRET efficiency may represent the structural flexibility associated with an intermediate state in the DNA binding and cleavage process, unrelated to the cleaved product." I missed how long the authors measured the FRET transitions for after adding to the TIRF. Are these transitions still observed in the complexes that are cleaved? Or do they terminate on the time scale of cleavage. If so, then what is the FRET state or distribution of FRET states? If not, that suggests that the product complex is not very different from the substrate-bound one and merits comment.

Page 12 (middle para 1): ", although the cleaved product does not occur unless the zipped conformation is stable." Stable is too vague here. This is about lifetimes of intermediates and how they affect downstream steps- the data indicate that the zipped conformation is on-pathway to the cleavage-competent state.

Page 12 (middle) "it is possible that Cas9 may sense the zipped conformation, triggering structural rearrangement of the nuclease domain". This doesn't make sense. Cas9 undergoes a conformational change into the zipped conformation that may reorient the nuclease domains for cleavage. There is no "sensing" involved.

2) The authors need to add details to figures and methods:

Figure 1e: What is the source of the dotted lines? Are these splines or fit to a kinetic equation. I think the authors should attempt curve fits to simple one and two-step kinetic models estimate the rate constants for cleavage. Arbitrary curves should not be used at all, or at least stated as being arbitrary. If the curves are fit to equations, then the equations should be given with the parameter values. There are also no error bars. What was n ?

Likewise for Figure 1D- the CD differences are small enough that multiple experiments with statistics are probably in order.

Page 20 Online methods "molecules) after making corrections for photo-bleaching and/or urea-mediated denaturation of the DNA duplex". What were the magnitudes of the corrections? What values or range of values were used?

3) The authors could be more specific in some of their statements:

Page 3(bottom) "However, the mechanistic details of the role of gRNAs in Cas9 nuclease activity are still unclear." This vague statement could be more specific about what is or is not yet known. Otherwise it doesn't convey much.

Page 5 (top) "..and crRNA added later". Perhaps "immediately afterwards" would be clearer.

Page 5(Para 1): "which time the kinetic curve of DNA cleavage was entirely saturated under our experimental conditions." This statement presumably means cleavage is complete or maximized over time, but this way of saying it is unclear. How about "the time required to achieve maximum cleavage."?

Page 5 (middle): "As a control, however, the cleavage efficiency without pre-incubation..." This is unclear. Is the "full complex" meaning 20 minute preincubation with both cr and tracr RNAs, while "without pre-incubation" means mixing them together and adding them directly to the TIRF? This is unclear.

Page 6 (Para 1) " The result, given in Fig. 1e,indicating that the thermal energy at 37 °C is sufficient for overcoming the conformational rearrangement barrier." For a biophysicist this is a funny way to say it as it implies a threshold, which of course is not true - this is just Arrhenius behavior. However, the authors could attempt to estimate the bounds of an approximate rate constant and ΔH from the data.

Final Sentence: "Cas9. Furthermore, this newly unveiled role of the kinetics and dynamics of the R-loop structure of Cas9:gRNA:DNA could help to quantify the off-target effects and enable more rational design for minimizing off-target effects in genome editing." I think that the authors could be more explicit in what they mean here.

4) The authors could be more quantitative in some statements:

Page 4 (middle of the page) "... the ratio of Cy5 to Cy3 decreased significantly, indicating that DNA cleavage had occurred". To read this does not give a sense of how much cleavage occurred. Also, the time resolution is not made clear in the text.

Page 5 (middle) "Temperature-controlled circular dichroism (CD) spectra suggests that the energy-consuming transformation from activated Cas9 to deCas9 is associated with a large-scale conformational change". Maybe "Temperature-dependent spectra". Also, the authors don't quantify the CD spectrum and so it is unclear how much structural change there is or why they use the term "large-scale". Maybe it would be useful to state it in terms of % loss of secondary structure or number of amino acids unfolded.

Page 8 (Para 1): "These results suggest that the low-FRET state in our measurement represents the PAM-proximal bound complex (termed the 'open conformation') in which crRNA partially hybridizes with the dsDNA before the complete formation of the R-loop." It would be nice to know, based on the crystal structures what the distances are likely to be.

Page 9 (middle) "we interpret that the shortened nucleotide results in a low binding energy ..." How low? Low compared to what?'

Page 10 (bottom) "high-FRET state, despite sufficient complementarity at the PAM-distal..." What is meant by "sufficient"? Sufficient for what?

5) The authors may want to make some statements clearer:

Page 9 (lower middle) "may exhibit enhanced target DNA specificity against off-target DNA strands" Confusing wording- how about "...enhanced discrimination against..."?

Page 10 (middle) "Furthermore, a comparison of the kineticsshowed that the transition to the open conformation resulted in a slower rate of DNA cleavage". Saying it this way suggests the transition (which occurs all of the time) is what is responsible for the behavior I think the authors are trying to say "showed that the greater proportion of time spent in the open conformation resulted in....".

Page 10 (middle) "that Cas9-mediated cleavage event is" should be "Cas9-mediated cleavage events are"

Page 10 (bottom) "In a similar way to the above cleavage experiments, " maybe should be "In agreement with the enhanced cleavage of the bubble substrate,"

Page 11 (top) " suggesting that the formation of the zipped conformation, in addition to the complementarity of nucleotides at the PAM-distal end, is required for the stability of the entire RNA-DNA heteroduplex". This seems backwards to me: formation of the zipped conformation REQUIRES complementarity through the last 10 nt. The results suggest that this is what drives the conformational change. Am I missing something?

Page 11(top) "..., the conformational dynamics of the R-loop structure is seen to reflect the enzymatic activity of the Cas9 complex as well as the progress of RNA-DNA base pairing." This seems too vague. Be explicit: the transition between the two states involves both R-loop expansion as well as catalytic activation- the two processes are coupled.

Reviewer #3 (Remarks to the Author):

Manuscript #NCOMMS-16-05762

Structural roles of guide RNAs in the nuclease activity of Cas9 endonuclease

By Lim et al.

Summary

Cas9 is a RNA-guided endonuclease that forms the basis of a powerful genome editing technology. This study has aimed to investigate the mechanistic roles of the guide RNAs (crRNA and tracrRNA) in programming Cas9 for DNA binding and cleavage. Using single-molecule spectroscopy, the authors provide evidence for a new catalytically inactive form of Cas9 that appears to be thermodynamically more stable than active Cas9 in the absence of tracrRNA. Single-molecule FRET experiments reveal the existence of distinct conformations of the target DNA-bound Cas9-guide RNA complex whose relative populations and dwell times are dependent on the extent of complementarity between the crRNA guide and the target DNA.

Overall, this is a very interesting and timely study of the biophysical properties of Cas9. While extensive X-ray crystallographic studies have revealed the basic molecular architecture and mechanisms of Cas9-mediated DNA binding and cleavage, our understanding of its molecular mechanism is far from complete. Recent studies by the Doudna group have revealed that the conformational state of the HNH domain in Cas9 controls the DNA cleavage activity of Cas9 and that this is dependent on the extent of complementarity between the guide RNA and the target DNA strand (Sternberg et al., Nature 2015).

This study provides important complementary insights into the conformational activation of the Cas9 endonuclease. As such, it sheds light not only on the basic molecular mechanism of the enzyme but also could facilitate the prediction of off-target activities of Cas9 and catalyze further development of next-generation variants of the basic CRISPR-Cas9 technology to reduce off-target effects. I find that the study is appropriate in its focus and scope for a manuscript in Nature Communications and would recommend it for publication provided that the following issues can be addressed by the authors.

Major comments:

1. While the study reveals that Cas9 converts to a deactivated state upon incubation at elevated temperatures in the absence of tracrRNA, the biological significance of this state, if any, is not clear. Non-specific RNA and DNA binding by SpCas9 has been observed previously (Jinek et al., Science 2012) and this study merely reiterates the fact. It would be interesting to establish whether this is a general property of other Cas9 endonucleases or a specific feature of SpCas9; however, this is clearly beyond the scope of the manuscript.

2. The study asserts that in the absence of tracrRNA, Cas9 undergoes a thermodynamically favourable transition that converts it to a catalytically inactive state. However, since the read-out of enzymatic activity necessarily involves adding the guide RNAs and applying the Cas9 to a DNA target, it is also possible that the observed "deactivation" of Cas9 results from inhibition due to non-specific binding of the crRNA or target DNA to an otherwise functional Cas9, as has been suggested before (Jinek et al., Science 2012). This would effectively trap the protein in dead-end complexes that may simply not bind the target DNA at all or bind the DNA in a way that does not lead to DNA cleavage. To distinguish these possibilities, it would be paramount to modify the experimental method such that both DNA binding and cleavage could be measured in a single experiment - e.g. by using a crRNA or tracrRNA labeled with one fluorophore and a target DNA labeled in the distal part of the duplex.

3. The authors claim that the inactive state of Cas9 corresponds to a novel conformation and support their claim with circular dichroism data that shows a decrease in ellipticity in the 210-220 nm range. However, another plausible explanation for the observed behavior would simply be unfolding of the polypeptide chain or a phase transition that would not involve a well-defined, specific conformational change, for example due to temperature-dependent changes in protein solubility. It would be highly informative if the authors could provide additional biophysical data to characterize the deactivated state, for example thermal melting curves and/or dynamic or static light scattering.

4. As far as I can discern, the target binding and cleavage data presented in Fig. 4a and b are obtained from two different experiments utilizing different RNA/DNA labeling schemes, but this is not explicitly stated in the text. It would be helpful if the authors could provide more details about the experimental setup in the text. Also, it is not clear what end time point was used for quantifying the fraction of DNA cleaved. One would assume that this is at least the same as the equilibrium time point used for the binding experiment, but it would be helpful if this was also clarified in the text.

5. Based on Figures 2d, 4a and 4b, having a substantial population of the target-bound Cas9 complexes in the high FRET state correlates with a high fraction of DNA cleavage. Is this also reflected the kinetics of DNA cleavage observed with the 17-nt crRNA?

6. On page 10, when discussing Supplementary Fig. 7, the authors conclude that "...transition to the open conformation resulted in a slower rate of DNA cleavage". I find this formulation rather awkward. Comparison of the kinetics of DNA cleavage for the wild-type and bubble DNA substrates suggests that formation of the high-FRET state is facilitated by mismatches between the complementary and non-complementary DNA strands in the PAM-distal part of the target duplex. This would suggest that once target DNA is bound by the Cas9-RNA complex, DNA melting in the PAM-distal part of the DNA duplex and formation of the RNA-DNA hybrid is the rate determining step. This would also argue that the low-FRET state is in fact a bona fide intermediate on the reaction trajectory rather than a "byroad" that reduces the population of the zipped conformation, as suggested in the legend text for Supplementary Figure 7.

Minor comments:

The manuscript could benefit from additional editing to improve the clarity of the text. I do not find it helpful to have a special name for the deactivated Cas9 (deCas9). I also find the terms used to describe the two states of the DNA-bound complex ("zipped" and "open") somewhat confusing. In several instances, the authors use rather imprecise or awkward expressions. For example:

1. Page 5:

"In contrast, Cas9 pre-incubated in the absence of tracrRNA..., which suggests that the tracrRNA plays a crucial role in the doorstep process to the subsequent binding of crRNA..."

Consider revising the sentence to

"..., which suggests that the tracrRNA plays a crucial role in facilitating crRNA binding and DNA cleavage by Cas9".

2. Page 6:

"Taken together, these results suggest that the conformational structure of deCas9 is thermodynamically more stable and severely locked at lower temperatures owing to the large refolding barrier."

What does "severely locked" mean?

3. Page 10:

"These results reveal that the open conformation is not involved in DNA cleavage and suggest that Cas9-mediate cleavage event is initiated only if a stable R-loop forms".

Consider revising the sentence to:

"These results reveal that the open conformation is incompatible with DNA cleavage and suggest that DNA cleavage is initiated only if a stable R-loop forms".

We would like to thank the three referees for the very constructive and positive reviews on our manuscript (#NCOMMS-16-05762). We have carefully studied the issues raised by the referees and revised our manuscript accordingly. We have run new experiments, added new supplementary data, and revised the main text according to the referee comments.

In their comments, all three reviewers expressed their dissatisfaction with our term 'deactive Cas9', and we changed it to 'inactive Cas9' or 'structurally inactive Cas9'.

Please find our detailed responses below. The referee comments are shown in black and our responses are in blue.

Reviewer #1 (Remarks to the Author):

The article by Lim et al. is on an important system that has been very actively investigated in the last few years. The CRISPR-Cas9 system provides a relatively simple yet a potent and versatile method of genome editing. Its great potential and the broad interest in CRISPR-Cas9 would clearly make studies on this system appropriate for Nature Communications. Another positive aspect of this work is that there are very few single molecule studies on this system, and in this respect the current manuscript provides a different method of probing the CRISPR-Cas9 system.

In this study, the authors propose that Cas9 can attain an inactive (deactive as they call it) conformation at elevated temperatures (37C) which is thermodynamically more stable than the active form. If slowly cooled to room temperature (25C) the protein remains in this inactive form however, if incubated with gRNA for extended periods of time the active conformation can be attained. The authors conclude that gRNA helps Cas9 attain the active conformation and prevents it from transitioning to the inactive conformation. Circular Dichroism (CD) data is provided demonstrating a large structural change when temperature is increased from 25C to 37C, which is proposed to be due to the transition from the active to the inactive form. In addition, the authors have manipulated the distal end of RNA and the complementarity between the DNA strands to demonstrate different types of kinetics and steady state distributions for DNA-crRNA complex.

However, I do not think significant new information that is not already observed in other studies is reported in this study. I also think the authors could have made a better case of relating their findings to the results obtained in recent studies. For example, the large structural transition that is observed in the circular dichroism spectra is not compared with the findings of crystal structure data or biochemical data (also obtained at elevated temperature as in this work). Similarly, the data in Figure 4 could have been discussed in the context of a number of other studies that point to a core region of 10-12 bp in the vicinity of PAM being essential for stable complex formation. Because such points were not clarified or their originality not clearly demonstrated, I do not think the authors have provided a convincing case for the significance of their study.

Below are my specific comments about this study:

1- I have some questions about how the "Fraction of Cleaved Product" is determined. I think this point should clearly be described as a significant part of the conclusions of this study depends on how these fractions are determined. This would also make a more convincing case for robustness and reliability of the assay. In this context, I think at least Supplementary Figure 1a should be within the body of the paper as it is essential to understand the assay. I have grouped my questions on this aspect below:

i) In Supplementary Figure 1, the authors demonstrate their reasoning for determining the fraction of cleaved product. In that assay, Cy5 is on the DNA strand that is not attached via biotin-neutravidin linker. Therefore, cleaving of the DNA should release this strand and result in a reduction in the number of Cy5 spots on the surface. The number of Cy3 spots, on the other hand, should not decrease in this

assay as Cy3-labeled strand is attached to the surface via a biotin-neutravidin linker.

However, a problem with this assay is that the cleaved product is not released by Cas9-gRNA complex, as demonstrated in other studies as well. Therefore, the authors have utilized 7M Urea to show that the DNA is actually cleaved. However, 7M Urea is a fairly strong denaturant and it is not clear if the reduction in the number of Cy5 spots is due to cleaving of the Target dsDNA or simply due to its unwinding by the Urea. For the latter to occur, only the part of DNA that does not hybridize with crRNA and remains in double stranded form after Cas9-gRNA binds needs to be unwound. The authors mention that they quickly flush out the urea from the sample chamber, but they have not actually shown if this process does not actually unwind the relevant part of the DNA construct. Unwinding of the Target dsDNA by Urea would in principle also give rise to removal of the Cy5 spot from the surface, regardless of whether it is cleaved or not. A potential control might be to repeat the same measurement with dCas9, the protein that can bind to dsDNA but cannot cleave it, and show that the number of Cy5 spots does not change before and after urea treatment. For this control, it is essential to make sure that crRNA hybridizes DNA, so that part of DNA is not in dsDNA form.

First of all, since the fact that DNA cleavage can be observed by using 7M urea is already reported by a previous study (Sternberg et al., Nature 2014), we would rather leave our Supplementary Figure 1a where it is now rather than making it part of the main text. Now, to address the referee comments regarding the 7M urea treatment, we performed a new control experiment to determine whether the treatment actually denatures dsDNA that forms heteroduplex with crRNA. We repeated the same cleavage assay with dCas9 to verify the effect of 7M urea solution on dsDNA or biotin-neutravidin denaturation after RNA-DNA heteroduplex was formed with dCas9. The result, shown in the following figure (also added to Supplementary Fig. 1), clearly shows that no significant DNA cleavage was observed.

Author Response Figure 1. Control experiment for the cleavage assay with dCas9

ii) Do the authors have to add Urea every time they determine the fraction of cleaved products or was it done in the Supplementary Figure 1 just to illustrate a point? If they do not use Urea every time, and the cleaved product is not released (at least within a short time), how do they determine what fraction of Cy5 molecules have left? I guess the alternative is to wait for a long enough time, but it is not clear if that is what was done.

We added 7M urea solution every time we determine the fraction of cleaved products.

iii) Another question I have about this assay is about the quantification aspect of it. Do the authors count the number of Cy5 spots before and after adding the Cas9-gRNA complex and take the ratio of the two to determine the fraction of cleaved products? Or do they consider the ratio of number of Cy5 spots to number of Cy3 spots? Do the authors follow a particular protocol to take into account photobleaching

of Cy5 molecules during the imaging?

We counted the number of Gaussian-fitted spots from measured images and then calculated the ratio of number of Cy3 to that of Cy5 before and after adding the Cas9-gRNA complex with the 7M urea treatment. (Please refer to the online methods section.) The effect of photobleaching should be negligible since our imaging time (5 s for actual data acquisition) is much shorter than the photobleaching time (~ 170 s) of Cy5 (see the data denoted “w/o gRNA” in Fig. 1b)).

2-The authors propose that Cas9 transitions into a deactive state at 37C as this is a thermodynamically more stable state. After it attains this inactive form, the cleavage activity is proposed to significantly reduce. As an evidence for this structural transition, they demonstrate CD data that show a change in ellipticity when temperature is raised from 25C to 37C but not when it is slowly cooled down from 37C to 25C. They interpret this as the deactive form being more stable and the protein being essentially trapped in this form at 25C as it does not have enough thermal energy to make the transition.

I have a comment and some questions about these observations and interpretation.

i) Jiang, Doudna et al. (Science 2015) also observe a large structural rearrangement of helical domain 3 upon binding to guide RNA. This ~65 Å conformational change is considered to be driven by binding of Cas9 to guide RNA. The authors of the current manuscript also propose that interactions with tracrRNA help Cas9 to transition into the active conformation. Can the authors comment on whether they consider these two conformational changes related? I am not sure if a clean enough CD spectra can be obtained but measuring the spectrum of Cas9 at 25C alone and with gRNA (after subtracting the gRNA spectrum from the complex) might be a way to check if these two large structural changes are related. Similarly, is it possible that the temperature induced shift the authors observe in CD spectra might be due to this type of structural transition? As this is a major conclusion of this study, I think the authors should elaborate on how this result fits in the broader context of various studies performed on this system.

We believe that the previously observed rearrangement of the helical domain 3 of Cas9 upon gRNA binding is not related with the conformational transition we report here between the activated and deactivated forms of Cas9. According to our result, sufficient pre-incubation of Cas9 without tracrRNA is required at an elevated temperature (e.g. 37 °C) to induce the transition toward deCas9, but no such condition was provided in the sample preparation described in the previous papers reporting crystal structures of Cas9 complexes, e.g. apo-Cas9 (Jinek et al, Science 2014) and Cas9-gRNA complex (Jiang et al, Science 2015). This indicates that the previously reported crystal structure of apo-Csa9 represent the activated form of Cas9, not the deactivated one. Moreover, our result indicates that the deactivated state cannot interact with tracrRNA to form a catalytically active Cas9-gRNA complex (Supplementary Fig. 2), implying that the Cas9 structure in the crystal structure of Cas9-gRNA complex is not associated with the deactivated state. Therefore, we believe that the previously reported rearrangement of the helical domain 3 involves a structural transition of the activated Cas9 interacting with the gRNA, whereas the deactivated Cas9 reported in the present study is a newly discovered state, not related to the previously reported crystal structure of apo-Cas9 or Cas9-gRNA complex. As for the referee comment on the CD spectra, we appreciate the experimental suggestion regarding the structural transition of deCas9, but we are afraid that “subtracting the gRNA spectrum from the complex” may not be detectable in our CD measurements since the gRNA forms a significant secondary structure in the Cas9-gRNA complex, which may very well be considerably different from the structure of free gRNA.

ii) In Sternberg et al. Nature 2014 article (Doudna-Greene collaboration), the authors compare cleaving activity of Cas9 at 25C and 37C, and they do not observe a significant difference (Extended Data Figure 5c-d) at these two temperatures. In addition, in both cases maximum "Fraction of Cleaved DNA" is reached within 1 minute, without the need for extensive recovery time. How would the authors reconcile

their proposed model of deactive state with these results?

In the Nature article, as the reviewer mentioned, the authors compared DNA cleaving activity of Cas9 at different DNA incubation temperatures and did not observe a significant difference. It is to be noted that all Cas9–RNA complexes in the Nature article were reconstituted (pre-incubated) at 37 °C, whereas we changed the temperature in the pre-incubation process, not in the incubation process of DNA (See Fig 1c).

iii) The first 25 C data in CD (black curve in Fig. 1D) is presumably in the active conformation since it is different from that in 37 C. If the deactive form is of higher stability, and hence lower free-energy and the Cas9 protein in these measurement does not interact with any of the RNA strands, why is Cas9 in the active form in the first 25C measurement? Shouldn't we expect it to be in the deactive form since that state has a lower free-energy?

We propose that inactive Cas9 is thermodynamically more stable than activated Cas9, but kinetically unfavorable. It is to be noted that sufficient energy such as to be provided at 37 °C to result in the conformational change to inactive Cas9 is not applied anywhere in our sample preparation step.

3- In Figure 2b, the authors observe a low and a high FRET peak which they attribute to partial (open conformation) and complete (zipped conformation) formation of R-loop respectively. In the wild type construct, there are frequent transitions between the two states. When a mismatch is introduced between DNA and crRNA at the distal end, the FRET naturally drops as RNA cannot form a heteroduplex with DNA at that end. Depending on how long of a mismatch is introduced, the FRET should drop accordingly and the authors have introduced a long enough mismatch to match the 0.27 peak. However, in this model I would expect a more continuous distribution of FRET states as R-loop expansion would be expected to follow a gradual path. However, in Fig. 2b-c, two well defined and separate peaks are observed as if there are two well defined steps in the process. For example, Fig. 2C shows that even when the system comes down from the high FRET state, it does not go to some intermediate FRET level but back to that particular low-FRET state. The authors could consider plotting a transition density plot of such transitions and demonstrate whether the transitions take place between two discrete states or from a broad range of low and intermediate states to a well defined high FRET state. Such an analysis would also benefit from higher time resolution.

Following the referee comment, we made a transition density plot from the single-molecule FRET traces at higher temporal resolution (0.03 s), which is shown below. (To determine the transition of FRET states, we used hidden Markov modelling based on variational Bayesian expectation maximization (Bronson et al., 2009).) Only two states are observed with no significant intermediate state.

Author Response Figure 2. Transition density plot

Along these lines, when the authors create a bubble-DNA that cannot form dsDNA but can hybridize with crRNA over an extended length, only the high FRET state is formed and no transitions are observed. These data suggest that the observed transitions are due to competition between dsDNA formation and

DNA-crRNA hybridization at the distal end. When dsDNA is formed, the Cy5 on RNA is pushed away from the Cy3 and when DNA-crRNA are hybridized the two fluorophores approach each other. In the case of bubble DNA, this competition is eliminated since dsDNA cannot form by design and hence there is nothing to compete with the DNA-crRNA hybridization. It is also worth pointing that the way the fluorophores are placed in this construct, only the dynamics at the distal end could be observed as anything that happens in the vicinity of PAM site would not influence the FRET levels in this construct. I think this is the model the authors also propose, but I am not sure since they made a bit more general statements in terms of the relative motions between target DNA and crRNA at the distal end, which by design has to be the case. Maybe some clarification could help here.

To clarify and further verify our model, we carried out additional experiments by employing new dye-labeling schemes in the DNA and crRNA constructs to visualize the motion of RNA-DNA heteroduplex at the PAM-proximal region during R-loop expansion. A set of DNA/crRNA constructs with different positions (denoted by base's number from PAM site) of donor/acceptor were prepared, with the acceptor fluorophore (Cy5, red) placed on crRNA and the donor fluorophore (Cy3, green) located on the complementary DNA, as shown in the figure below. We note that the change in the inter-fluorophore distance gets smaller as the positions of the fluorophores get closer to the PAM site. Especially, the FRET between Cy3 (#4 on complementary DNA strand) and Cy5 (#10 on crRNA) clearly shows no transitions (bottom panel). Our data supports our model that the pairing of PAM-proximal region exhibits stable binding after PAM recognition, as in the Argonaute-microRNA system (Bartel et al., Cell 2009). Several structural studies on Cas9 (Jinek et al., Science 2014 and Nishimasu et al., Cell 2014) also support our observation of the stable conformation in the PAM-proximal seed. We believe these results support our conclusion that the structural transition occurs only in the PAM-distal end while a stable heteroduplex persists in the PAM-proximal region.

Author Response Figure 3. FRET histograms for other positions of FRET dye-pair

I agree with the authors in interpreting Fig. 3 however, I also think that this result is expected based on how the fluorophores are placed on the DNA. By using a 17 nt crRNA, the competition between dsDNA formation and DNA-crRNA hybridization is biased towards dsDNA formation. Therefore, the low

FRET population is expected to dominate in this case. The values for the FRET peaks are also expected since in the zipped state the fluorophore is now 3 bp further in the truncated construct (17 nt cr-RNA) compared to the full construct (20 nt cr-RNA), therefore the high FRET state is expected to reduce. The low FRET state is not expected to show as much variation as the separation is shortened by ~3 nt. The shortening of the dwell time in high FRET state is also explained by essentially the bias towards dsDNA formation, and weakening of DNA-crRNA hybridization. I think these observations could be made without any reference to Cas9 system, they are just nucleic acids interacting with each other. The more interesting aspect would probably be how such variations influence the function of the CRISPR-Cas9 complex and its dsDNA cleaving or sequence recognition activities, which I don't think are addressed.

First of all, we agree with the referee that what we observed in Fig. 3 can be explained by the positions of the fluorophores on the DNA and considering the competition between dsDNA formation and DNA-crRNA hybridization. As for its implications on the function of the CRISPR-Cas9 complex and its dsDNA cleaving or sequence recognition activities, after noting the enhanced target-specificity of the Cas9 complex with the truncated crRNA (Fu et al., 2014), what we intended was to interpret our result in view of our main idea in this work that the crRNA complementarity to target DNA (i.e. the interaction of RNA-DNA heteroduplex) affects the enzymatic activities of Cas9 complex. Regarding the dsDNA cleaving activity, we pointed out that the difference in the population of high-FRET state results in a different rate constant in the DNA cleavage of Cas9 complex. Furthermore, we also demonstrated that in the truncated case the decrease in the population of high-FRET state causes a decrease in the rate constant for DNA cleavage activity of the Cas9 complex (See Supplementary Fig. 7, Author Response Fig. 5 and our reply to reviewer #3).

4-As far as I understand, all images were taken at an integration time of 0.5 s, yet the authors quote errors as small as 0.05 s in their dwell time measurements (e.g. Supplementary Figure 6b). This does not look very convincing, especially considering that the fitting function is essentially dominated by several bins in the histogram. In the same data, the characteristic dwell times based on exponential fits are 0.4 s for high FRET state and 1.1 s for low FRET state. Given a 0.5 s integration time, such differences are probably marginally distinguishable. The sample FRET trace shown in Figure 6a is probably a good example of how challenging it would be to be able to distinguish between these two numbers because most transitions to high FRET state only have 1-2 data points, which makes it very challenging to distinguish them from fluctuations in the data. Similarly, it would be quite challenging to measure a dwell time of 0.4 s, as most dwell times will have one data point in them. Also, there are a large number of events that do not obviously belong to high or low FRET states. The authors mention "an appropriate threshold" was used to sort high and low FRET values but a threshold might be challenging to implement when there is only 1-2 data points to work with. A large number of groups have consistently obtained clean smFRET data with Cy3-Cy5 with 50-100 ms integration time using a similar camera. Did the authors attempt taking data with higher time resolution to make these results more convincing?

We very much appreciate this comment, which is an absolutely correct point that we sorely overlooked. We repeated the entire experiments and measured the kinetics at a higher time resolution of 0.03 s. We used these new data to replace Fig. 2c, 3b and Supplementary Fig. 5, 6. As for "an appropriate threshold" we mentioned, to determine the FRET states and each transition in single-molecule time trajectories, we used hidden Markov modelling (Bronson et al., 2009).

5- A minor comment about the image in Supplementary Figure 1: 30 bp is a very long distance to observe significant FRET between Cy3-Cy5, yet the Cy5 channel appears very bright. At 30 bp end-to-end distance would be >8 nm. Given a Cy3-Cy5 Forster radius of ~5.6 nm, the FRET would be expected to be around 0.1. If the contrast is not adjusted to make the Cy5 spots brighter, the FRET would appear to be > 0.5, even before incubation at which point we should not expect any protein-induced enhancement or unwinding of part of the duplex. Is there something I am missing here?

We apologize for the unclear description that led to the misunderstanding. For all cleavage assays in our work, we did not intend FRET interaction between two dyes. The Cy5 molecules are independently excited by 633 nm He-Ne laser; We added this statement to the online methods section.

6- Can the authors also quote sigma or sigma of mean for the peak positions? There are several peak positions varying between 0.76-0.83 or 0.27-0.29, and it is not clear if these differences are significant.

High and low FRET efficiencies with 20-nt crRNA are 0.83 and 0.27 respectively, while high and low FRET efficiencies with the truncated 17-nt crRNA are 0.76 and 0.29 respectively. We interpreted that the values for FRET state are different since the distance between two fluorophores is 3-bp farther in the truncated RNA-DNA duplex (17-nt crRNA) compared to the full RNA-DNA duplex (20-nt crRNA), thus the high FRET state showing a reduced FRET value. The low FRET state, however, shows a slightly increased FRET value, which might be due to the formation of the secondary structure between the unpaired single-stranded crRNA at the PAM-distal end and HNH, RuvC domains of Cas9 protein, as recently reported by cryo-EM (Jiang et al., Science 2016).

Reviewer #2 (Remarks to the Author):

Using single molecule FRET and TIRF imaging, Lim et al claim to have characterized two conformational transitions in Cas9/gRNA complexes controlled by the RNAs. In their model, tracrRNA promotes the formation of an "active" conformation early in the assembly pathway, which they demonstrated indirectly using kinetic analysis. The crRNA controls/promotes a transition between "open" (initial) and "zipped" (cleavage active) states of the Cas9/gRNA/targetDNA complex after binding, which they demonstrated using single molecule FRET between the 5'-end (PAM-distal) of the crRNA and the non-hybrid DNA target strand.

The data suggest that in the absence of tracrRNA, Cas9 transitions (on the minute time scale at 37°C) to an inactive conformation that can be induced to slowly reactivate (also on the minute time scale) by tracrRNA. The crRNA-induced transitions occur on the second time scale and are modulated by RNA-DNA complementarity at the PAM-distal end of the heteroduplex.

The authors posit that the crRNA-induced conformational change coincides with R-loop expansion from an initial binding state that samples the PAM-proximal end of the target DNA to the cleavage competent state where the entire crRNA is hybridized with the target strand. They suggest that the time distribution spent between the open and zipped states is determined by PAM-distal complementarity which in turn controls cleavage rate and target specificity.

This paper is the first to investigate in real time conformational changes on Cas9 assembly and target selection. While earlier single molecule work identified distinct PAM-proximal and PAM-distal dependent binding steps and postulated R-loop expansion as the associated structural change (Szczelkun et al 2014), this work provides data that support this connection, provides equilibrium distributions and time scales, and provides direct evidence linking this change to downstream cleavage. Likewise, open and zipped conformations were postulated from crystal structures and provided support for interpreting the FRET data. Understanding the role of tracrRNA in active complex assembly might be useful in maximizing in vitro assembly reactions or trouble-shooting in vivo ones. The crRNA controlled transition provides mechanistic insights into how Cas9/gRNA sample dsDNA for target site selection. This work should be of interest to researchers dissecting the structural and kinetic mechanisms of target site selection could certainly, as the authors suggest, lead to design of enhanced specificity variants. The claims are convincing given the coincidence of the FRET data and other previously published works. The method is internally controlled as all measurements were performed in the context of the TIRF set-up.

Experiments that would strengthen these claims are those that would further test the predictions of their model (observing FRET behavior of Cas9 mutations designed to stabilize one form or the other, and how these affect specificity) seem beyond the scope of a Communications format. The authors appear to have covered most of the relevant literature (although I am not a Cas9 "expert"). One recent publication they might cite and/or discuss involves enhanced specificity mutants gained by destabilizing the PAM-proximal interactions on the non-cleaving strand which retain high cleavage activity even though they bind less well (Slaymaker et al Science 351, 84 (2016)).

We followed the referee's suggestion and added the reference in our revised manuscript.

Another addition which might be useful, is quantification of transition rate constants for the tracrRNA-dependent slow step as well as rate constants for the cleavage. It is also possible to examine the ΔH via a temperature-dependence series, but this would be some work and perhaps beyond what is required for a Communications paper.

There are some minor scientific and writing issues:

1) The authors might want to address these scientific points:

Page 7 (bottom) "Rather, these two different values of the FRET efficiency may represent the structural flexibility associated with an intermediate state in the DNA binding and cleavage process, unrelated to the cleaved product." I missed how long the authors measured the FRET transitions for after adding to the TIRF. Are these transitions still observed in the complexes that are cleaved? Or do they terminate on the time scale of cleavage. If so, then what is the FRET state or distribution of FRET states? If not, that suggests that the product complex is not very different from the substrate-bound one and merits comment.

The FRET traces were recorded immediately following the injection of Cas9-gRNA complex onto the immobilized DNA intermittently for up to ~ 30 min after the injection. Although the fact that the dwell-time ratio of the low-to-high FRET states (Supplementary Fig. 5) does not match the population ratio of the FRET histogram (Fig. 2b) may imply that the cleaved products remain in high FRET states, the transition still occurs considerably even after the time scale of cleavage. Due to this ambiguity, we cannot conclude whether or not the transition occurs in the cleaved product unless the cleavage event is directly monitored. We are now trying to employ a new experimental scheme that enables real-time observation of the cleavage event for further study.

Page 12 (middle para 1): ", although the cleaved product does not occur unless the zipped conformation is stable." Stable is too vague here. This is about lifetimes of intermediates and how they affect downstream steps- the data indicate that the zipped conformation is on-pathway to the cleavage-competent state.

We changed the sentence to (page 12) "... although the cleavage does not occur unless the zipped conformation is formed".

Page 12 (middle) "it is possible that Cas9 may sense the zipped conformation, triggering structural rearrangement of the nuclease domain". This doesn't make sense. Cas9 undergoes a conformational change into the zipped conformation that may reorient the nuclease domains for cleavage. There is no "sensing" involved.

We changed the sentence to (page 12) "it is possible that Cas9 undergoes a conformational change into the zipped conformation that may trigger structural rearrangement of the nuclease domain".

2) The authors need to add details to figures and methods:

Figure 1e: What is the source of the dotted lines? Are these splines or fit to a kinetic equation. I think

the authors should attempt curve fits to simple one and two-step kinetic models estimate the rate constants for cleavage. Arbitrary curves should not be used at all, or at least stated as being arbitrary. If the curves are fit to equations, then the equations should be given with the parameter values. There are also no error bars. What was n? Likewise for Figure 1D- the CD differences are small enough that multiple experiments with statistics are probably in order.

We thank the reviewer for raising this valid point. We agree with the referee comment and deleted the dotted line. Although we found there is a rate-determining step involving a reaction intermediate regarding the initial deCas9 kinetics, the mechanistic details are still unclear and we cannot build an appropriate kinetic model at this stage. Following additional referee comments, we added more data points and error bars in Fig. 1e.

Page 20 Online methods "molecules) after making corrections for photo-bleaching and/or urea-mediated denaturation of the DNA duplex". What were the magnitudes of the corrections? What values or range of values were used?

We apologize for the incorrect description. There was no additional correction. It was the ratiometric evaluation (i.e., [the number of Cy3 molecule] divided by [the number of Cy5 molecules]) that was to exclude the effect of photo-bleaching and/or urea-mediated denaturation. We changed the sentence to (page 20) "... (number of Cy3 molecules)/(number of Cy5 molecules) after making consideration for photo-bleaching and/or urea-mediated denaturation of the DNA duplex". For more details on the quantification process, please also see our reply to reviewer #1.

3) The authors could be more specific in some of their statements:

Page 3(bottom) "However, the mechanistic details of the role of gRNAs in Cas9 nuclease activity are still unclear." This vague statement could be more specific about what is or is not yet known. Otherwise it doesn't convey much.

We changed the sentence to (page 3) "However, the mechanistic details of the independent roles of the two gRNAs in Cas9 nuclease activation are still unclear", and also the subsequent sentence to "The goal of this study is to investigate and unravel the molecular mechanism of the two gRNAs separately on Cas9:gRNA complexation and their conformational changes associated with the nuclease activity".

Page 5 (top) "...and crRNA added later". Perhaps "immediately afterwards" would be clearer.

We changed the sentence to (page 5) "... and crRNA added immediately afterwards".

Page 5(Para 1): "which time the kinetic curve of DNA cleavage was entirely saturated under our experimental conditions." This statement presumably means cleavage is complete or maximized over time, but this way of saying it is unclear. How about "the time required to achieve maximum cleavage."?

We changed the sentence to (page 5) "... the time required to achieve maximum cleavage under our experimental conditions".

Page 5 (middle): "As a control, however, the cleavage efficiency without pre-incubation..." This is unclear. Is the "full complex" meaning 20 minute preincubation with both cr and tracr RNAs, while "without pre-incubation" means mixing them together and adding them directly to the TIRF? This is unclear.

Yes, "without pre-incubation" means direct injection of the mixture with apo-Cas9, crRNA and tracrRNA not passing through pre-incubation at 37 °C. To avoid confusion, we changed the sentence to (page 5) "As a control, however, the cleavage efficiency for the direct injection of the mixture with apo-Cas9, crRNA, and tracrRNA ('without pre-incubation', Fig. 1c, gray bar)...".

Page 6 (Para 1) " The result, given in Fig. 1e,indicating that the thermal energy at 37 °C is sufficient for overcoming the conformational rearrangement barrier." For a biophysicist this is a funny way to say it as it implies a threshold, which of course is not true - this is just Arrhenius behavior. However, the authors could attempt to estimate the bounds of an approximate rate constant and ΔH from the data.

We apologize for the misleading description. We did not intend to use '37 °C' as a specific threshold in the conformational rearrangement. To avoid confusion, we changed the sentence to (page 6) "The result, given in Fig. 1e,indicating that the thermal energy at the elevated temperature allows more frequent occurrence of the conformational rearrangement."

Final Sentence: "Cas9. Furthermore, this newly unveiled role of the kinetics and dynamics of the R-loop structure of Cas9:gRNA:DNA could help to quantify the off-target effects and enable more rational design for minimizing off-target effects in genome editing." I think that the authors could be more explicit in what they mean here.

To be more explicit, we changed the final sentence to "Furthermore, this newly unveiled role of the kinetics and dynamics of the R-loop structure of Cas9:gRNA:DNA could allow prediction of off-target sites quantitatively based on the calculation of the stability of the crRNA-DNA interface, thereby enabling more efficient target design in genome editing".

4) The authors could be more quantitative in some statements:

Page 4 (middle of the page) "... the ratio of Cy5 to Cy3 decreased significantly, indicating that DNA cleavage had occurred". To read this does not give a sense of how much cleavage occurred. Also, the time resolution is not made clear in the text.

We changed the sentence to (page 4) "... the ratio of Cy5 to Cy3 decreased significantly; approximately 80% of target DNAs were cleaved in 5 min".

Page 5 (middle) "Temperature-controlled circular dichroism (CD) spectra suggests that the energy-consuming transformation from activated Cas9 to deCas9 is associated with a large-scale conformational change". Maybe "Temperature-dependent spectra". Also, the authors don't quantify the CD spectrum and so it is unclear how much structural change there is or why they use the term "large-scale". Maybe it would be useful to state it in terms of % loss of secondary structure or number of amino acids unfolded.

We changed the sentence to (page 5) "Temperature-dependent circular dichroism (CD) spectra suggest that the energy-consuming transformation from active Cas9 to inactive Cas9 is associated with a conformational change". The structural difference between active and inactive forms of Cas9 cannot be quantified from our CD spectra because both of them contribute to the CD data even at 37 °C. At higher temperatures, denaturation of Cas9 starts to occur so that the CD spectrum of deCas9 alone could not be obtained. (Please also refer to our reply to reviewer #3.)

Page 8 (Para 1): "These results suggest that the low-FRET state in our measurement represents the PAM-proximal bound complex (termed the 'open conformation') in which crRNA partially hybridizes with the dsDNA before the complete formation of the R-loop." It would be nice to know, based on the crystal structures what the distances are likely to be.

The FRET efficiency of the open conformation (0.27) amounts to an inter-dye distance of 70 Å with $R_0 = 56$ Å, but unfortunately we are not able to verify this value since there is no available crystal structure representing the PAM-proximal bound complex.

Page 9 (middle) "we interpret that the shortened nucleotide results in a low binding energy ..." How low? Low compared to what?"

We changed the sentence to (page 9) "we interpret that the shortened nucleotide results in a lower

binding energy than the binding energy with 20-nt crRNA at the PAM-distal end of the crRNA-DNA interface, destabilizing the zipped conformation”.

Page 10 (bottom) "high-FRET state, despite sufficient complementarity at the PAM-distal..." What is meant by "sufficient"? Sufficient for what?

RNA-DNA heteroduplex possessing the mismatched bases at 10-11 position has a melting temperature of 59.91 °C (from the theoretical calculation based on the article by Dumousseau M et al., 2012). We use the term “sufficient” to point out the high melting temperature or thermal stability of the RNA-DNA heteroduplex in the complex at such incubating temperature and salt concentrations. Following the referee comment, however, we changed the sentence to (page 11) “... high-FRET state, despite the complementarity of 9 nucleotides at the PAM-distal end...”.

5) The authors may want to make some statements clearer:

Page 9 (lower middle) "may exhibit enhanced target DNA specificity against off-target DNA strands" Confusing wording- how about "...enhanced discrimination against..."?

Page 10 (middle) "Furthermore, a comparison of the kineticsshowed that the transition to the open conformation resulted in a slower rate of DNA cleavage". Saying it this way suggests the transition (which occurs all of the time) is what is responsible for the behavior I think the authors are trying to say "showed that the greater proportion of time spent in the open conformation resulted in....".

Page 10 (middle) "that Cas9-mediated cleavage event is" should be "Cas9-mediated cleavage events are"

Page 10 (bottom) "In a similar way to the above cleavage experiments, " maybe should be "In agreement with the enhanced cleavage of the bubble substrate,"

We made all efforts to reflect these referee comments in our revised manuscript.

Page 11 (top) " suggesting that the formation of the zipped conformation, in addition to the complementarity of nucleotides at the PAM-distal end, is required for the stability of the entire RNA-DNA heteroduplex". This seems backwards to me: formation of the zipped conformation REQUIRES complementarity through the last 10 nt. The results suggest that this is what drives the conformational change. Am I missing something?

We apologize our lapse and corrected the sentence to (page 11): “ suggesting that the formation of the zipped conformation, in addition to the complementarity of nucleotides at the PAM-distal end, requires the stability of the entire RNA-DNA heteroduplex”.

Page 11(top) "..., the conformational dynamics of the R-loop structure is seen to reflect the enzymatic activity of the Cas9 complex as well as the progress of RNA-DNA base pairing." This seems too vague. Be explicit: the transition between the two states involves both R-loop expansion as well as catalytic activation- the two processes are coupled.

We changed the sentence to (page 12) “...the FRET transition between the two states involves both R-loop expansion as well as the enzymatic activation of Cas9 complex”.

Reviewer #3 (Remarks to the Author):

Summary

Cas9 is a RNA-guided endonuclease that forms the basis of a powerful genome editing technology. This study has aimed to investigate the mechanistic roles of the guide RNAs (crRNA and tracrRNA) in programming Cas9 for DNA binding and cleavage. Using single-molecule spectroscopy, the authors provide evidence for a new catalytically inactive form of Cas9 that appears to be thermodynamically more stable than active Cas9 in the absence of tracrRNA. Single-molecule FRET experiments reveal the existence of distinct conformations of the target DNA-bound Cas9-guide RNA complex whose relative populations and dwell times are dependent on the extent of complementarity between the crRNA guide and the target DNA.

Overall, this is a very interesting and timely study of the biophysical properties of Cas9. While extensive X-ray crystallographic studies have revealed the basic molecular architecture and mechanisms of Cas9-mediated DNA binding and cleavage, our understanding of its molecular mechanism is far from complete. Recent studies by the Doudna group have revealed that the conformational state of the HNH domain in Cas9 controls the DNA cleavage activity of Cas9 and that this is dependent on the extent of complementarity between the guide RNA and the target DNA strand (Sternberg et al., Nature 2015).

This study provides important complementary insights into the conformational activation of the Cas9 endonuclease. As such, it sheds light not only on the basic molecular mechanism of the enzyme but also could facilitate the prediction of off-target activities of Cas9 and catalyze further development of next-generation variants of the basic CRISPR-Cas9 technology to reduce off-target effects. I find that the study is appropriate in its focus and scope for a manuscript in Nature Communications and would recommend it for publication provided that the following issues can be addressed by the authors.

Major comments:

1. While the study reveals that Cas9 converts to a deactivated state upon incubation at elevated temperatures in the absence of tracrRNA, the biological significance of this state, if any, is not clear. Non-specific RNA and DNA binding by SpCas9 has been observed previously (Jinek et al., Science 2012) and this study merely reiterates the fact. It would be interesting to establish whether this is a general property of other Cas9 endonucleases or a specific feature of SpCas9; however, this is clearly beyond the scope of the manuscript.

We appreciate this suggestion. We are currently investigating the conversion to a deactivated conformational state by incubating with other CRISPR endonuclease (e.g. CRISPR-Cpf1 endonuclease) to address the generality and biological significance of this state.

2. The study asserts that in the absence of tracrRNA, Cas9 undergoes a thermodynamically favourable transition that converts it to a catalytically inactive state. However, since the read-out of enzymatic activity necessarily involves adding the guide RNAs and applying the Cas9 to a DNA target, it is also possible that the observed "deactivation" of Cas9 results from inhibition due to non-specific binding of the crRNA or target DNA to an otherwise functional Cas9, as has been suggested before (Jinek et al., Science 2012). This would effectively trap the protein in dead-end complexes that may simply not bind the target DNA at all or bind the DNA in a way that does not lead to DNA cleavage. To distinguish these possibilities, it would be paramount to modify the experimental method such that both DNA binding and cleavage could be measured in a single experiment - e.g. by using a crRNA or tracrRNA labeled with one fluorophore and a target DNA labeled in the distal part of the duplex.

Since the two states of Cas9 (active and inactive form) are differentiated by the means of DNA cleavage activity, all components of the Cas9 complex (i.e., Cas9, crRNA, tracrRNA, and DNA) should be mixed at the end to determine the cleavage efficiency in our experimental scheme (Fig. 1a), which is the point the referee raises regarding the contribution of non-specific binding of crRNA or DNA. Our work shows, however, that apo-Cas9 pre-incubated alone at the elevated temperature (e.g. 37 °C) transforms into the inactive state (Fig. 1b (iii)). In both cases where apo-Cas9 was pre-incubated alone and 'without pre-incubated' (for the precise meaning of this terminology, please refer to our revised manuscript and our

reply to reviewer #2), Cas9 species would experience the same amount of inhibition from non-specific binding during the same period of 5-min DNA incubation. If the non-specific binding of crRNA or DNA to Cas9 is the origin for the deactivation, the DNA cleavage efficiency of the two cases should be the same. Therefore, although we cannot exclude the possibility that the inactive state has non-specific interaction/binding with crRNA or DNA, it is clear that the inhibition from the non-specific interaction is not the main culprit for the deactivation.

On the other hand, we have attempted to monitor both DNA binding and cleavage to distinguish the behavior of the cleaved product with that of non-cleaved intermediate, but practically could not perform the experiment because the Cas9 complex does not show the liberation from the cleaved dsDNA.

3. The authors claim that the inactive state of Cas9 corresponds to a novel conformation and support their claim with circular dichroism data that shows a decrease in ellipticity in the 210-220 nm range. However, another plausible explanation for the observed behavior would simply be unfolding of the polypeptide chain or a phase transition that would not involve a well-defined, specific conformational change, for example due to temperature-dependent changes in protein solubility. It would be highly informative if the authors could provide additional biophysical data to characterize the deactivated state, for example thermal melting curves and/or dynamic or static light scattering.

We very much appreciate this suggestion. As the major goal of this paper is to explore the role of gRNAs in Cas9 nuclease activation, we have focused on the role of tracrRNA which reactivates the inactive form of Cas9 to the active one, not the inactive state per se. Nonetheless, to address the referee comments, we ran new experiments and obtained thermal melting curve for apo-Cas9 by monitoring the shift of intrinsic tryptophan fluorescence at the emission wavelengths of 330 nm (F330) and 350 nm (F350). The result, given in the following figure, shows the change in the tryptophan fluorescence of apo-Cas9 upon thermal unfolding. From the first derivative of the plot of the fluorescence ratio F330/F350, we unambiguously determined the melting temperature (T(m)) of apo-Cas9 to be 48.5 °C.

Author Response Figure 4. Thermal melting curve

$$f_D = \frac{(y_N - y)}{(y_N - y_D)} \quad (1)$$

f_D : fractional denaturation

y_N : observed value of characteristic of native states

y_D : observed value of characteristic of denatured states

From equation (1), we obtained only 2% of the denatured fraction at 37 °C, suggesting the effects of an unfolded polypeptide chain or phase transition at 37 °C are negligible. Therefore, we believe that this

thermal melting curve would relieve the reviewer's concern about the inactive state being an unfolded or phase-transition state.

4. As far as I can discern, the target binding and cleavage data presented in Fig. 4a and b are obtained from two different experiments utilizing different RNA/DNA labeling schemes, but this is not explicitly stated in the text. It would be helpful if the authors could provide more details about the experimental setup in the text. Also, it is not clear what end time point was used for quantifying the fraction of DNA cleaved. One would assume that this is at least the same as the equilibrium time point used for the binding experiment, but it would be helpful if this was also clarified in the text.

We revised our manuscript according to the referee comments.

5. Based on Figures 2d, 4a and 4b, having a substantial population of the target-bound Cas9 complexes in the high FRET state correlates with a high fraction of DNA cleavage. Is this also reflected the kinetics of DNA cleavage observed with the 17-nt crRNA?

Yes. We measured the cleavage rate with the 17-nt crRNA and compared its rate constant to those measured with other DNA constructs (see the table below). The observed rate constant with 17-nt crRNA is much smaller than the rest. This is in agreement with our result that only the population of Cas9 complex in high-FRET state (zipped conformation) is associated with DNA cleavage. See the figure and table below.

Author Response Figure 5.

	Bubble DNA	Wild-type DNA (bona fide)	Wild-type DNA with 17-nt crRNA
Observed rate constant (min^{-1})	1.59	0.58	0.07

Author Response Table 1. Rate constants

6. On page 10, when discussing Supplementary Fig. 7, the authors conclude that "...transition to the open conformation resulted in a slower rate of DNA cleavage". I find this formulation rather awkward. Comparison of the kinetics of DNA cleavage for the wild-type and bubble DNA substrates suggests that formation of the high-FRET state is facilitated by mismatches between the complementary and non-complementary DNA strands in the PAM-distal part of the target duplex. This would suggest that once target DNA is bound by the Cas9-RNA complex, DNA melting in the PAM-distal part of the DNA duplex and formation of the RNA-DNA hybrid is the rate determining step. This would also argue that the low-FRET state is in fact a bona fide intermediate on the reaction trajectory rather than a "byroad" that reduces the population of the zipped conformation, as suggested in the legend text for

Supplementary Figure 7.

We agree with the point raised by the referee and changed the term ‘byroad’ to ‘a bona fide intermediate’ in the legend text for Supplementary Figure 7. We also changed the sentence in consideration of reviewer #2’s suggestion: “...the greater proportion of time spent in the open conformation resulted in...”, and added the interpretation that “once target DNA is bound by the Cas9-RNA complex, DNA melting in the PAM-distal part of the DNA duplex and formation of the RNA-DNA hybrid is the rate determining step”.

Minor comments:

The manuscript could benefit from additional editing to improve the clarity of the text. I do not find it helpful to have a special name for the deactivated Cas9 (deCas9). I also find the terms used to describe the two states of the DNA-bound complex ("zipped" and "open") somewhat confusing. In several instances, the authors use rather imprecise or awkward expressions. For example:

Following the referee suggestion, we changed the term ‘deCas9’ to ‘structurally inactive Cas9’ or ‘inactive Cas9’ in our revised manuscript.

1. Page 5:

"In contrast, Cas9 pre-incubated in the absence of tracrRNA..., which suggests that the tracrRNA plays a crucial role in the doorstep process to the subsequent binding of crRNA..."

Consider revising the sentence to

"..., which suggests that the tracrRNA plays a crucial role in facilitating crRNA binding and DNA cleavage by Cas9".

We adopted the referee comments and changed the sentence accordingly.

2. Page 6:

"Taken together, these results suggest that the conformational structure of deCas9 is thermodynamically more stable and severely locked at lower temperatures owing to the large refolding barrier."

What does "severely locked" mean?

We changed the phrase ‘severely locked’ to ‘kinetically trapped’.

3. Page 10:

"These results reveal that the open conformation is not involved in DNA cleavage and suggest that Cas9-mediate cleavage event is initiated only if a stable R-loop forms".

Consider revising the sentence to:

"These results reveal that the open conformation is incompatible with DNA cleavage and suggest that DNA cleavage is initiated only if a stable R-loop forms".

We adopted the referee comments and changed the sentence accordingly.

Reviewer #1 (Remarks to the Author):

The authors have satisfactorily answered the issues I raised in the earlier review. I think they have made a serious effort to answer all referee comments by taking new data, performing new analyses, and making corrections and clarifications in the manuscript where appropriate. I would recommend publication of this revised version.

Reviewer #2 (Remarks to the Author):

Rereview of “Structural roles of guide RNAs in the nuclease activity of Cas9 endonuclease”

I reread the revised paper and looked at the responses to my previous comments. The authors did a good job of addressing my concerns and writing more clearly. While I think this paper is generally ready for publication, there are still a couple of wordings the authors should reconsider that are technically incorrect with regards to thermodynamics:

1) Abstract and elsewhere: “...a new conformation of structurally inactive Cas9 that is thermodynamically more preferable than active Cas9.” What is “active” Cas9? From my view it means complexed with gRNAs “cleavage competent”. However, the authors here mean an apo conformation that is competent to assemble into a cleavage-competent complex. The authors should consider defining “active” early in the paper, or using the term “active apo-Cas9”.

2) Page 7, top: “...Cas9-tracrRNA interaction prevents Cas9 from undergoing a thermodynamically favorable conformational change to inactive Cas9.” This is not really a technically correct statement- it “prevents it” by generating a lower energy species in its complex with tracrRNA- using LeChatlier’s principle. It would be more correct to say it skews the equilibrium away from the inactive conformation by lowering the energy of the active one.

Reviewer #3 (Remarks to the Author):

The re-submitted manuscript by Lim et al., together with the accompanying response to the reviewers' comments, now provides additional insights. Overall, the manuscript has been substantially improved and I appreciate the authors' efforts to address my concerns raised in the first round of review. I only have a few minor comments at this point.

1. The biological significance of the inactivated state that SpCas9 is proposed to convert to at elevated temperatures is still not clear and not discussed by the authors. After all, this protein needs to function in a human pathogen at 37°C. It might be argued that *in vivo*, SpCas9 actually does not exist in a true apo state for significant periods of time since it is likely to be stably bound to a tracrRNA. This would be consistent with the notion that tracrRNA binding prevents SpCas9 inactivation and the previously reported non-specific DNA binding by SpCas9 observed in the absence of tracrRNA. I think that this is a point that should be raised in the manuscript.

2. The data shown in Figure 1c shows temperature-dependent inactivation of SpCas9 in the presence of the crRNA guide. Has the same thermally-induced behavior been observed with apo-SpCas9? It has recently been reported that the solubility of the binary SpCas9-guide RNA complex is dramatically reduced under low salt conditions (similar to the buffer conditions used in this manuscript) compared to apo-SpCas9 (see Burger, A., et al. (2016). Maximizing mutagenesis with solubilized CRISPR-Cas9 ribonucleoprotein complexes. *Development* 143, 2025–2037). In their response to my previous comments, the authors have provided thermal denaturation curves for apo-SpCas9 but they have not tested the behavior of the SpCas9-crRNA complex. So although only 2% of apo-SpCas9 may be unfolded at 37°C, a much greater fraction of the SpCas9-crRNA complex might be inactive (due to reduced solubility and aggregation) at the same temperature, explaining the observed discrepancy. It is also possible that the inactivation of the SpCas9-crRNA complex is due to a solubility issue while the inactivation behavior of apo-SpCas9 is due to a different effect and the authors are conflating the two.

3. On page 5, in the sentence "As a control, however, the cleavage efficiency for the direct injection of the mixture with apo-Cas9, crRNA, and tracrRNA ('without pre-incubation', Fig. 1c, gray bar) retained nearly 100% of the cleavage activity of the 'full complex'" – what is meant by the 'full complex'? SpCas9-crRNA-tracrRNA or SpCas9-crRNA-tracrRNA-target DNA?

4. On page 6, the authors state: "The presence of the lag phase indicates that there is a rate-determining step involving a reaction intermediate, which could be attributed to the slow and unfavorable conformational rearrangement from inactive Cas9 to active Cas9." This would imply that the reactivation of Cas9 occurs first and is followed by tracrRNA binding, which stabilizes Cas9 and activates it for DNA binding. However, it is also plausible that tracrRNA binding and Cas9 reactivation are concerted and result in a conformational rearrangement, as observed in the structure of the SpCas9-sgRNA binary complex (Jiang, F., et al. (2015). A Cas9-guide RNA complex preorganized for

target DNA recognition. *Science* 348, 1477–1481). Although possibly slow due to a rate-limiting step, this process is nevertheless thermodynamically highly favourable.

5. I find the term “structurally inactivated Cas9” unsubstantiated, since the authors’ data does not convincingly show that the inactivation is due to SpCas9 adopting a well-defined autoinhibited structure. I suggest that the term “inactivated Cas9” is used consistently throughout the manuscript.

6. Fig. 1E – is the mark in the middle of the graph an error?

Reviewer #1 (Remarks to the Author):

The authors have satisfactorily answered the issues I raised in the earlier review. I think they have made a serious effort to answer all referee comments by taking new data, performing new analyses, and making corrections and clarifications in the manuscript where appropriate. I would recommend publication of this revised version.

Thank you very much for your approving comments.

Reviewer #2 (Remarks to the Author):

Re-review of “Structural roles of guide RNAs in the nuclease activity of Cas9 endonuclease”

I reread the revised paper and looked at the responses to my previous comments. The authors did a good job of addressing my concerns and writing more clearly. While I think this paper is generally ready for publication, there are still a couple of wordings the authors should reconsider that are technically incorrect with regards to thermodynamics:

Thank you very much for your encouraging comments. Our responses to the additional comments are as follows:

1) Abstract and elsewhere: “... a new conformation of structurally inactive Cas9 that is thermodynamically more preferable than active Cas9.” What is “active” Cas9? From my view it means complexed with gRNAs “cleavage competent”. However, the authors here mean an apo conformation that is competent to assemble into a cleavage-competent complex. The authors should consider defining “active” early in the paper, or using the term “active apo-Cas9”.

Following the reviewer’s suggestion, we have changed the phrase “active Cas9” to “active apo-Cas9”.

2) Page 7, top: “... Cas9-tracrRNA interaction prevents Cas9 from undergoing a thermodynamically favorable conformational change to inactive Cas9.” This is not really a technically correct statement – it “prevents it” by generating a lower energy species in its complex with tracrRNA – using LeChatelier’s principle. It would be more correct to say it skews the equilibrium away from the inactive conformation by lowering the energy of the active one.

This is of course a thermodynamically correct description. We changed the relevant part to “... Cas9-tracrRNA interaction shifts the equilibrium away from the inactive conformation, the thermodynamically favorable state in the absence of tracrRNA, by allowing a lower energy state for the active one.”

Reviewer #3 (Remarks to the Author):

The re-submitted manuscript by Lim et al., together with the accompanying response to the reviewers’ comments, now provides additional insights. Overall, the manuscript has been substantially improved and I appreciate the authors’ efforts to address my concerns raised in the first round of review. I only have a few minor comments at this point.

Thank you very much for your encouraging comments. Our responses to the additional comments are as follows:

1. The biological significance of the inactivated state that SpCas9 is proposed to convert to at elevated temperatures is still not clear and not discussed by the authors. After all, this protein needs to function in a human pathogen at 37oC. It might be argued that in vivo, SpCas9 actually does not exist in a true

apo state for significant periods of time since it is likely to be stably bound to a tracrRNA. This would be consistent with the notion that tracrRNA binding prevents SpCas9 inactivation and the previously reported non-specific DNA binding by SpCas9 observed in the absence of tracrRNA. I think that this is a point that should be raised in the manuscript.

The reviewer's daring yet highly plausible suggestion is indeed one that we wanted to make a key point of our paper but fell short of proving by experiment. In full agreement with the reviewer's suggestion, we suspect that most SpCas9 exist as spCas9–tracrRNA–crRNA or spCas9–sgRNA complex rather than inactive apo-Cas9 alone in human cells at 37 °C so that gene editing is successfully induced. However, the detailed step-by-step procedure for the formation of spCas9-gRNA complex *in vivo* still remains to be elucidated. We hope to be able to address the biological significance of inactive spCas9 through further *in vivo* study. Considering the current limitation of our study, we added the phrase in the Discussion section (page 11) “although its significance *in vivo* remains to be addressed by further studies”.

2. The data shown in Figure 1c shows temperature-dependent inactivation of SpCas9 in the presence of the crRNA guide. Has the same thermally-induced behavior been observed with apo-SpCas9? It has recently been reported that the solubility of the binary SpCas9-guide RNA complex is dramatically reduced under low salt conditions (similar to the buffer conditions used in this manuscript) compared to apo-SpCas9 (see Burger, A., et al. (2016). Maximizing mutagenesis with solubilized CRISPR-Cas9 ribonucleoprotein complexes. *Development* 143, 2025–2037). In their response to my previous comments, the authors have provided thermal denaturation curves for apo-SpCas9 but they have not tested the behavior of the SpCas9-crRNA complex. So although only 2% of apo-SpCas9 may be unfolded at 37°C, a much greater fraction of the SpCas9-crRNA complex might be inactive (due to reduced solubility and aggregation) at the same temperature, explaining the observed discrepancy. It is also possible that the inactivation of the SpCas9-crRNA complex is due to a solubility issue while the inactivation behavior of apo-SpCas9 is due to a different effect and the authors are conflating the two.

The same temperature-dependency has been observed in the case of apo-Cas9, implying that the thermally induced transition of apo-Cas9 occurs in the absence of both tracrRNA and crRNA.

Author Response Figure 1. Thermal melting curve of Cas9-crRNA

In the *Development* paper by Burger *et al.*, it is the ‘Cas9-sgRNA’ complex that shows a low solubility under low salt conditions, but not the ‘Cas9-crRNA binary complex’ whose existence has not been identified so far. It might be possible that transient, non-specific interaction between Cas9 and crRNA induces aggregation or reduces solubility of Cas9 species. However, in the thermal melting curve of Cas9 in the presence of crRNA (Author Response Fig. 1), we cannot find significant denaturation or a phase transition at 37 °C.

Furthermore, the full Cas9-gRNA complex cleaved target DNA very well during the same period of 5-min DNA incubation (‘full complex’ in Fig. 1b), indicating that the poor solubility of the Cas9-gRNA complex does not lead to a significant decrease in the cleavage efficiency under our experimental conditions. Therefore, although a small fraction of Cas9 species may exist as an insoluble protein and/or complex at 37 °C due to the phase transition or aggregation, it is clear that the insoluble forms are not the major factor for the inactivation that dramatically reduces the cleavage efficiency from ~ 80% to ~ 20%.

3. On page 5, in the sentence “As a control, however, the cleavage efficiency for the direct injection of the mixture with apo-Cas9, crRNA, and tracrRNA (‘without pre-incubation’, Fig. 1c, gray bar) retained nearly 100% of the cleavage activity of the ‘full complex’” – what is meant by the ‘full complex’? SpCas9-crRNA-tracrRNA or SpCas9-crRNA-tracrRNA-target DNA?

The ‘full complex’ means Cas9-gRNA complex as we have already defined it in the main text (page 5): “apo-Cas9 pre-incubated at 37 °C for 20 min with both gRNAs (full complex)”.

4. On page 6, the authors state: “The presence of the lag phase indicates that there is a rate-determining step involving a reaction intermediate, which could be attributed to the slow and unfavorable conformational rearrangement from inactive Cas9 to active Cas9.” This would imply that the reactivation of Cas9 occurs first and is followed by tracrRNA binding, which stabilizes Cas9 and activates it for DNA binding. However, it is also plausible that tracrRNA binding and Cas9 reactivation are concerted and result in a conformational rearrangement, as observed in the structure of the SpCas9-sgRNA binary complex (Jiang, F., et al. (2015). A Cas9-guide RNA complex preorganized for target DNA recognition. *Science* 348, 1477–1481). Although possibly slow due to a rate-limiting step, this process is nevertheless thermodynamically highly favourable.

We very much appreciate this comment. As shown in Supplementary Fig. 2, tracrRNA does not directly interact with inactive Cas9 to form catalytically active Cas9-gRNA complex at 25 °C. However, it is still possible that the interaction between tracrRNA and inactive Cas9 is blocked by an energy barrier that is not overcome at 25 °C but becomes surmountable at 37 °C. Considering such a possibility, we changed the relevant sentence (in page 6) to “The presence of the lag phase indicates that there is a rate-determining step involving reaction intermediates, which could be attributed to the slow conformational rearrangement of inactive Cas9 involving tracrRNA”.

5. I find the term “structurally inactivated Cas9” unsubstantiated, since the authors’ data does not convincingly show that the inactivation is due to SpCas9 adopting a well-defined autoinhibited structure. I suggest that the term “inactivated Cas9” is used consistently throughout the manuscript. We revised our manuscript according to the referee comment.

6. Fig. 1E – is the mark in the middle of the graph an error?
Yes, we apologize for our lapse. We erased the mark from the figure.